# LAVA: A UNIFIED FRAMEWORK FOR FINETUNING LANGUAGE AND VISION MODELS

## ABSTRACT

LoRA and its variants have attracted considerable attention because of their abilities to tune a negligible number of parameters while achieving comparable downstream performance. This success is largely attributed to the intrinsic low-rank structure of model parameter spaces, which allows LoRA to train two projection matrices to project weights into a low-dimensional subspace and then map them back. However, it does not consider how to explore this low-rank subspace sufficiently and may lose the expression ability accordingly. Moreover, when using LoRA to tune convolution layers, a flatten operation is required to convert tensors into matrices. We argue that this will degrade the model's performance. In this paper, we address this issue from a general parameter sub-space perspective: we present a unified **L**anguage **A**nd **V**ision **A**daption finetuning framework (called **LAVA**). Specifically, we verify the existence of low-rank subspaces in convolution layers empirically and propose to parameterize the increment of both convolution kernels and matrices as sum of learnable rank-1 components. To improve training stability, we analyze the optimization dynamics of LoRA and incorporate orthogonal regularization into our parameterization, for which we give theoretical proof that it will help reduce the variance of the gradient. We conduct various experiments on different downstreaming tasks to validate LAVA's superiority. For example, when tuning LLaMA2-7b for commonsense tasks, the performance of our LAVA is **+1.9%** higher than that of LoRA. For metric depth estimation tasks, LAVA only tunes $\sim$1.5% of Depth-Anything$_{\text{large}}$ (335.3M), and achieves **+3.5%** $\delta_1$ accuracy against that of LoRA and **+5.6%** $\delta_1$ accuracy against that of SVDiff.

## 1 INTRODUCTION

Pre-trained large models (PLMs) have shown their remarkable abilities across wide domains (Wan et al., 2025) (Siméoni et al., 2025) (Guo et al., 2025). By scaling the number of parameters and size of data, models exhibit unprecedented generalization (Chen et al., 2025) abilities. To tailor these models into specific task domains and bypass the bottleneck of the accelerator's memory, parameter-efficient fine-tuning methods (PEFT) are proposed. Supported by intrinsic dimensionality theory that there exists a low-dimensional subspace that is important for downstream tasks (Aghajanyan et al., 2020), LoRA freezes the original weight matrix and tunes two newly introduced matrices in this low-rank space to significantly reduce memory cost.

However, we argue that LoRA and its variants fail to explore the low-rank subspace sufficiently and thus prevent the model from finding a better solution in this restricted subspace. Specifically, the optimization of the low-rank factors $A \in \mathbb{R}^{m \times r}$ and $B \in \mathbb{R}^{r \times n}$ is typically unconstrained. As a result, multiple columns of $A$ (or rows of $B$) can collapse into highly correlated directions, leading to dimension redundancy: many dimensions may be redundant and contain noise in these additional dimensions. This will waste parameter capacity, only to degrade the model's performance.

Additionally, in some vision tasks that require pixel-wise granularity (e.g., image generation (Podell et al., 2023), depth estimation (Bhat et al., 2023), and image inpainting (Ju et al., 2024)), finetuning convolution layers is still needed: it can refine the regional details of images. Researchers normally full fine-tune the convolution layer, which ignores the existence of intrinsic dimensions in the convolution kernel. On the other hand, existing PEFT methods focus on attention and neglect the performance tuning on convolution. A naive way is to flatten the convolution weights into matrices,

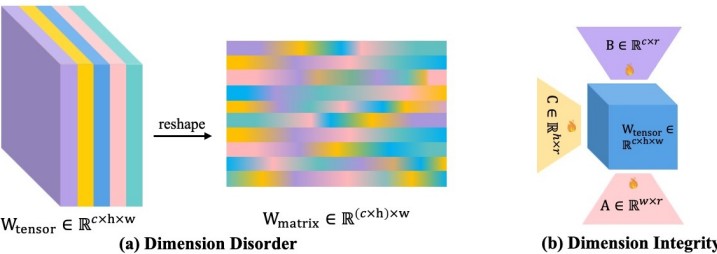

Figure 1: Comparisons between LoRA and our LAVA. (a): dimension disorder, which describes the dimension fusion when flattening tensors into matrices. (b): dimension integrity: which describes how our LAVA does to isolate the training for each dimension.

use LoRA to finetune them, and then restore them to the original tensor dimension. However, such a reshape-tune-restore training paradigm will inevitably force dimension disorder and disrupt the spatial encoding properties inherent in convolution: Fig. 1(a) shows why dimension disorder arises. Inspired by the analysis above, we naturally propose a question:

***Can we design a unified PEFT framework that adapts both attention- and convolution-based modules efficiently in low-rank subspaces across NLP and vision tasks?***

In this paper, we introduce **LAVA**, a unified **L**anguage **A**nd **V**ision **A**daptation framework. To our knowledge so far, we are the first to consider spatial properties in PEFT methods. The central idea is to think of matrices as 2D tensors and introduce generalized subspaces that keep dimensional integrity. We leverage dimension-isolated matrices and parameterize the incremental update into the sum of learnable rank-1 components. Fig. 1(b) illustrates the central idea of our LAVA. We show that when LAVA is applied to attention, it is a general form of LoRA. Moreover, to reduce dimension redundancy, we propose column orthogonal regularization and give theoretical proof that this helps reduce the variance of the gradient and stabilize the training. We validate LAVA across a wide range of tasks in various model architectures, and we evaluate our method in different model scales (from 125M in RoBERTa up to 7B in LLaMA2-7b). The experiment results show that LAVA consistently outperforms other PEFT methods across every domain: for example, in NLU, LAVA achieves **+1.2%** in average compared with LoRA; in commonsense reasoning tasks, LAVA is on average **+1.9%** higher than that of LoRA; and in depth estimation, the performance of tuning LAVA on Depth-Anything is **+3.5%** higher than that of LoRA.

Our main contributions can be summarized as follows:

- We propose a hypothesis that low-rank subspaces exist in convolution layers, and we empirically validate the existence of such subspaces.

- Based on the observation above, we propose LAVA, a unified language and vision adaptation method that introduces a general framework for tuning convolution and attention blocks.

- We theoretically prove that our proposed orthogonal regularization can stabilize the training of LAVA by reducing the variance of the gradient, and we empirically validate the robustness of different choices of hyperparameters and give insights on how to set hyperparameters accordingly.

- We conduct experiments to show that LAVA consistently surpasses other PEFT methods across various tasks and models: from LLaMA2-7b in commonsense reasoning to stable-diffusion-XL in text-to-image generation.

## 2 RELATED WORKS

We first give the preliminaries used throughout the paper. A matrix is denoted by an upper-case letter, e.g., $A$; $A_{ij}$ denotes the element of $A$ at the $i$-th row and $j$-th column. An $N$-th order tensor is denoted by a calligraphic letter, e.g., $\mathcal{X} \in \mathbb{R}^{I_1 \times I_2 \times \cdots \times I_N}$, where $N$ is the number of dimensions

of the tensor. $\circ$ denotes the outer-product operation, and we use python-style $A_{[:,r]}$ to slice the entire column $r$ of matrix $A$.

## 2.1 LOW-RANK ADAPTATION (LORA)

LoRA (Hu et al., 2022) is a common method in parameter-efficient finetuning. It uses low-rank matrices to approximate the real weight changes. When training, LoRA freezes the pre-trained weight matrix $W \in \mathbb{R}^{n \times m}$, and models the weight update $\Delta \in \mathbb{R}^{n \times m}$ with the multiplication of two smaller trainable matrices $A \in \mathbb{R}^{r \times m}$, $B \in \mathbb{R}^{n \times r}$, where $r \ll \min\{n, m\}$. We denote the input token of the current layer as $x \in \mathbb{R}^D$, the output as $y \in \mathbb{R}^D$, where $D$ is the hidden dimension of the model. Assuming $n = m = D$, the vanilla feed-forward pass is formulated as follows:

$$y = (W + BA)x. \tag{1}$$

The gradients with respect to $A$ and $B$ are:

$$\nabla_A L = B^\top \frac{\partial L}{\partial y} x^\top = B^\top (\nabla_W L), \nabla_B L = \frac{\partial L}{\partial y} x^\top A^\top = (\nabla_W L) A^\top, \tag{2}$$

where $L$ is the next-token prediction loss.

Recently, many variants of LoRA have been proposed to improve its performance from different perspectives. Inspired by weight normalization (Salimans & Kingma, 2016), DoRA (Liu et al., 2024) separates the direction and magnitude of the adapted matrix $BA$ apart: it introduces column-wise normalization on $BA$ so that it only controls the optimization direction, and a vector $m$ is trained to control the scale of each column. Hayou et al. (2024) analyzes the infinite-width setting and finds that A and B are not properly trained, and the asymmetry between them leads to inefficient feature learning. Thus, they propose to let the learning rate of the matrix $B$ be $\lambda$ times larger than that of $A$ to learn features sufficiently. Conv-Adapter (Chen et al., 2024) is a PEFT method designed for tuning convolution layers. It trains two smaller convolution layers, where one for aligning the size of the feature map with the output generated after the pre-trained convolution layer, and the other is a 1x1 convolution layer designed for controlling channel depth.

**Comparison against existing methods.** Many orthogonal-based finetuning methods have been proposed, while our method is based on a completely different idea. OLoRA (Büyükakyüz, 2024) is proposed to initialize $A$ and $B$ using the first $r$ QR-decomposition components of the pretrained matrix to speed up convergence, which means that it initializes $A$ by an orthonormal matrix. Our method focuses on training dynamics and approximates $A$ as an orthogonal matrix during training. However, OLoRA prioritizes initialization and merely thinks of the first step of optimization. Another relevant method is OFT (Qiu et al., 2023), which trains an orthogonal matrix and multiplies it onto the pre-trained matrix to preserve the energy. However, such multiplication-style tuning rotates the space, which is only applicable to easy or in-domain tasks, and it usually introduces more memory cost to store the entire transformation matrix. Compared with it, addition-style finetuning like LoRA allows more transformation freedom and can be used for more difficult tasks.

## 3 LOW-RANK SUBSPACE ANALYSIS

Drawing inspiration from Candecomp/Parafac (CP) decomposition, which represents the tensor as a sum of rank-1 tensor components while keeping the relations of the tensor dimensions, we introduce a new tensor increment analysis. Our analysis compares finetuning methods with and without reshaping operations and validates the existence of low-rank subspaces for convolution weights.

**Analysis Method**: Denote the weight of a convolution layer as $W_{tensor} \in \mathbb{R}^{c_{out} \times c_{in} \times h \times w}$, we compare two finetuning methods: reshape-involved (LoRA) and reshape-free. For reshape-involved methods, we reshape the original tensor into the shape $W' \in \mathbb{R}^{c_{out} \times (h \times c_{in} \times w)}$ and then use LoRA to tune the flattened matrix. On the other hand, reshape-free method parameterizes the update $\Delta W \in \mathbb{R}^{c_{out} \times c_{in} \times h \times w}$ into sum of rank-1 tensors. It keeps the dimensional integrity by introducing one matrix per tensor dimension. Suppose the rank of reshape-free way is $R$ and $X \in \mathbb{R}^{h \times R}$, $U \in \mathbb{R}^{c_{out} \times R}$, $Y \in \mathbb{R}^{w \times R}$ and $V \in \mathbb{R}^{c_{in} \times R}$, it directly reparameterizes the update tensor as the sum of $R$ outer product of four vectors:

$$\Delta W = \sum_{r=1}^{R} U_{[:,r]} \circ V_{[:,r]} \circ X_{[:,r]} \circ Y_{[:,r]}. \tag{3}$$

We conduct a toy experiment to compare the performances on ResNet18 (He et al., 2016), ResNet34 (He et al., 2016), VGG11 (Simonyan & Zisserman, 2014), AlexNet (Krizhevsky et al., 2012), GoogLeNet (Szegedy et al., 2015), and ConvNeXt (Liu et al., 2022) between methods with and without keeping dimensional integrity in low-rank subspaces. We inject trainable parameters into the convolution layer and finetune the model on the dataset CIFAR10 (Krizhevsky et al., 2009).

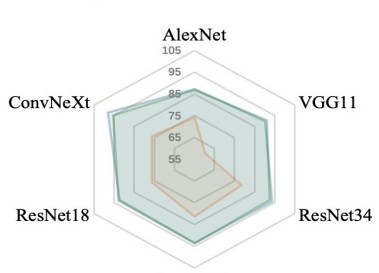

Figure 2: Comparisons in accuracy between reshape-free and reshape-involved methods.

Table 1: Trainable parameters of two reshape methods on six pre-trained models.

| Model | Reshape-involved | Reshape-free |
|---|---|---|
| AlexNet | 0.13 | 0.08 |
| VGG11 | 0.09 | 0.04 |
| ResNet34 | 1.05 | 0.25 |
| GoogLeNet | 2.32 | 1.66 |
| ResNet18 | 1.01 | 0.28 |
| ConvNeXt | 3.12 | 0.15 |

As is shown in Fig. 2 and Table 1, the reshape-free method outperforms that of reshape-involved by around 10% in all models using less than $\approx$50% trainable parameters. What's more, models finetuned by the reshape-free method even outperform full finetuning in ConvNeXt and VGG11, while the reshape-free method only tunes only $\sim$0.1% of total parameters. Detailed quantitative results can be found in Appendix A.4. The superiority gives us two insights:

- **Insight 1**: **It is important to preserve the dimension of tensors when finetuning the convolution layer**: maintaining the original multi-dimensional form ensures spatial correlations;
- **Insight 2**: **Convolution layers also exhibit intrinsic dimensionality during finetuning**: although convolution kernels are high-dimensional, only a small subset of directions in this space contribute effectively to downstream adaptation.

## 4 LAVA: GENERALIZED FINETUNING FRAMEWORK FOR BOTH VISION AND LANGUAGE TASKS

Drawing from the insights of our subspace analysis, in this section, we introduce a unified framework for finetuning language and vision models. Our method contains two important components: (i) low-rank subspace-based adaptation, which parameterizes the increment into a sum of rank-1 tensors; (ii) orthogonal regularization, which is applied on trainable blocks to stabilize the training.

### 4.1 REDUCING TENSOR TO MATRIX: GENERALIZED SUBSPACE-BASED ADAPTATION

When updating convolution or higher-order tensors, we propose to use Eq.( 3) to reparameterize the increment. This structure intrinsically isolates gradient computations and parameter updates per tensor dimension, as the partial derivatives $\frac{\partial L}{\partial (U_{[:,i]} \circ V_{[:,i]} \circ X_{[:,i]} \circ Y_{[:,i]})}$ depends solely on vectors related to $i$-th column. By representing $\Delta W$ as a sum of separable components, the training of $X$, $Y$, $U$, and $V$ becomes dimension-wise independent, reducing cross-dimensional interference.

As a matrix can be thought of as a 2D tensor, LAVA can be used in attention blocks as a general method of LoRA. For one matrix, the tensor dimension reduces to 2; Eq.( 3) can be reformulated into the following form:

$$W = W_0 + \Delta \approx W_0 + \sum_{r=1}^{R} U_{[:,r]} \circ V_{[:,r]} = W_0 + \sum_{r=1}^{R} U_{[:,r]}(V_{[:,r]})^\top = W_0 + UV^\top, \quad (4)$$

where $W \in \mathbb{R}^{m \times n}$, $U \in \mathbb{R}^{m \times R}$ and $V \in \mathbb{R}^{n \times R}$. Here, each $U_{[:,r]}(V_{[:,r]})^T$ is a rank-1 residual matrix to compensate for the gap between the expected and real incremental. And we notice that Eq.( 4) exactly matches the form of LoRA, meaning that LoRA is a special case of LAVA.

For the original tensor $\mathcal{X} \in \mathbb{R}^{c_{out} \times c_{in} \times h \times w}$, the memory required to store the tensor is $\mathcal{O}(c_{out} \times c_{in} \times w \times h)$. Assuming the rank is set to $r$, in reshape-involved LoRA, the memory is $\mathcal{O}((c_{out} + c_{in} \times h \times w) \times r)$, while in our method, the memory is $\mathcal{O}((c_{out} + c_{in} + w + h) \times r)$, which is typically far lower than that of reshape-involved LoRA.

## 4.2 Extending Matrix to Tensor: Column-Orthogonal Regularization Helps Stabilize The Training

Consider LAVA in matrix form: $\Delta \approx \sum_{r=1}^{R} U_{[:,r]}(V_{[:,r]})^\top$, each $V_{[:,r]}$ can be represented as a basis vector in the subspace $\mathcal{S} \subset \mathbb{R}^R$ and the corresponding $U_{[:,r]}$ determines the coordinate along this dimension. Unconstrained optimization may lead to an ill-conditioned subspace, whereas the orthogonal regularization in LAVA improves conditioning and can suppress redundant directions. We propose to apply column-orthogonal regularization $\lambda ||A^\top A - I||_F^2$ on $A$ during optimization to enforce orthogonality. In this situation, the gradient of $B$ keeps the same, and the gradient of $A$ becomes:

$$\nabla_A L = B^\top(\nabla_W L) + 4\lambda A(A^\top A - I). \quad (5)$$

As shown in Eq.( 3), in a convolution layer, there are four choices that one can take to apply orthogonal regularization on. To analyze the influence of different ways of applying regularization, we follow the setting shown in Section 3 and conduct the following experiment to pick the best way for regularization. From Fig. 11 in the Appendix, we empirically find that imposing orthogonality on $U$ makes a balance between accuracy and efficiency, which guides us to apply orthogonal regularization on $U$ (matrix corresponding to output channel).

The proposed orthogonal regularization indeed helps stabilize the training and reduce the variance in gradients, as shown in the following theorems:

**Theorem 1: Convergence rate under orthogonal regularization**. If the optimizer is SGD, denote the full gradient as $\bar{G}(t)$, gradient calculated on a mini-batch as $G(t)$, $G(t)$ satisfies: $\mathbb{E}[G(t)] = \bar{G}(t)$, $\mathbb{E}[||G(t) - \bar{G}(t)||_2^2] \leq \sigma^2$. Assuming $||\bar{G}(t)||_2 \leq \mu$, when the regularization strength term is set to $\lambda$, $A$ will get converged to orthogonal matrix $M$ ($M^\top M = I$) at the speed of $\mathcal{O}(\frac{\mu+\sigma}{\lambda})$.

It characterizes the convergence behavior of matrix $A$ towards an orthogonal structure, thereby offering a quantitative measure of how quickly the term $||A^\top A - I||_F$ diminishes during optimization. Please refer to Appendix A.2 for proof. Based on the theorem above, we can get the following proposition:

**Proposition 1: Upper bound of gradient variance**. Let $G = \nabla_W L \in \mathbb{R}^{m \times n}$, define the covariance matrix as $\Sigma_G = \mathbb{E}[(G - \mathbb{E}G)^\top(G - \mathbb{E}G)] \in \mathbb{R}^{n \times n}$, deviation error as $E_A = ||A^\top A - I||_2$. Suppose $tr(G) < +\infty$, then:

$$\mathrm{Var}(\nabla_B L) \leq (1 + E_A)tr(\Sigma_G).$$

As $E_A$ converges to 0 at the speed of $\mathcal{O}(\frac{\mu+\sigma}{\lambda})$, the coefficient term $(1 + E_A)$ will get closer to 1, and the variance of $\nabla_B L$ will fall into a small neighborhood region of FT variance $\Sigma_G$. In other words, penalizing $A$ with orthogonal regularization $||A^\top A - I||_F^2$ will reduce the variance of the gradient in $B$, and the algorithm is more likely to converge to a point with smaller gradient and a more stable oscillation radius. Detailed proof can be found in Appendix A.3.

Such regularization can be extended to tensor form as well. Suppose a 2D convolution layer $\mathcal{X} \in \mathbb{R}^{c_{out} \times c_{in} \times h \times w}$, LAVA decomposes the update of $\mathcal{X}$ into Eq.( 3). We apply column-orthogonal

regularization on $U$: $||U^\top U - I||_F^2$. Since LAVA decomposes the incremental into the sum of rank-1 tensors, adding orthogonal regularization will help LAVA to explore the rank-$R$ subspace more sufficiently while obeying dimensional integrity.

# 5 EXPERIMENTS AND RESULTS

In this section, we evaluate how orthogonal regularization benefits the training in attention blocks in NLP tasks. Then we switch to vision and multimodal tasks where convolution layers exist: we test whether LAVA is applicable for both convolution and matrix.

## 5.1 NATURAL LANGUAGE UNDERSTANDING

Firstly, we test LAVA on Natural Language Understanding (NLU) to showcase the effectiveness of applying orthogonal regularization in NLP tasks.

We finetune RoBERTa$_{\text{base}}$ (125.0M) on GLUE benchmark (Wang et al., 2018) following Wu et al. (2024a). In this experiment, we compare our method with baselines including Full-Finetuning, BitFit (Ben Zaken et al., 2022), Adapter (Houlsby et al., 2019), Adapter-FFN (Pfeiffer et al., 2020), LoRA (Hu et al., 2022), RED (Wu et al., 2024a), LoReFT (Wu et al., 2024b), and DeLoRA (Bini et al., 2025). Please refer to Appendix A.5 for detailed descriptions of the datasets, baseline, and hyperparameters.

Table 2: Comparisons of different methods finetuning RoBERTa on GLUE benchmark. The best result on each dataset is marked **bold**, and the second highest value is marked underline. Results are averaged on random seeds 42, 43, 44, 45, and 46.

| Method | # Params | MNLI | SST-2 | MRPC | CoLA | QNLI | QQP | RTE | STS-B | Avg. |
|---|---|---|---|---|---|---|---|---|---|---|
| FT | 125.0M | **87.3**$_{\pm0.34}$ | **94.4**$_{\pm0.96}$ | 87.9$_{\pm0.91}$ | 62.4$_{\pm3.29}$ | 92.5$_{\pm0.22}$ | **91.7**$_{\pm0.19}$ | 78.3$_{\pm3.20}$ | 90.6$_{\pm0.59}$ | 85.6 |
| Adapter | 0.4M | 87.0$_{\pm0.28}$ | 93.3$_{\pm0.40}$ | 88.4$_{\pm1.54}$ | 60.9$_{\pm3.09}$ | 92.5$_{\pm0.02}$ | 90.5$_{\pm0.08}$ | 69.8$_{\pm1.51}$ | 90.5$_{\pm0.35}$ | 85.0 |
| Adapter-FFN | 0.3M | 87.1$_{\pm0.10}$ | 93.0$_{\pm0.50}$ | 88.8$_{\pm1.38}$ | 58.5$_{\pm1.69}$ | 92.1$_{\pm0.28}$ | 90.2$_{\pm0.07}$ | 77.7$_{\pm1.93}$ | 90.4$_{\pm0.31}$ | 84.7 |
| BitFit | 0.1M | 84.7$_{\pm0.08}$ | 94.0$_{\pm0.87}$ | 88.1$_{\pm1.57}$ | 54.0$_{\pm3.07}$ | 91.0$_{\pm0.05}$ | 87.3$_{\pm0.02}$ | 69.8$_{\pm1.51}$ | 89.5$_{\pm0.35}$ | 82.3 |
| LoReFT | 0.02M | 83.1$_{\pm0.26}$ | 93.4$_{\pm0.64}$ | 89.2$_{\pm2.62}$ | 60.4$_{\pm2.60}$ | 91.2$_{\pm0.25}$ | 87.4$_{\pm0.23}$ | **79.0**$_{\pm2.76}$ | 90.0$_{\pm0.29}$ | 84.2 |
| RED | 0.02M | 83.9$_{\pm0.14}$ | 93.9$_{\pm0.31}$ | 89.2$_{\pm0.98}$ | 61.0$_{\pm2.96}$ | 90.7$_{\pm0.35}$ | 87.2$_{\pm0.17}$ | 78.0$_{\pm2.06}$ | 90.4$_{\pm0.32}$ | 84.3 |
| LoRA | 0.3M | 86.6$_{\pm0.23}$ | 93.9$_{\pm0.49}$ | 88.7$_{\pm0.76}$ | 59.7$_{\pm4.36}$ | 92.6$_{\pm0.10}$ | 90.4$_{\pm0.08}$ | 75.3$_{\pm2.79}$ | 90.3$_{\pm0.54}$ | 84.7 |
| DeLoRA | 0.3M | 87.2$_{\pm0.15}$ | 94.1$_{\pm0.70}$ | 89.0$_{\pm0.96}$ | **63.6**$_{\pm1.52}$ | 92.8$_{\pm0.51}$ | 90.1$_{\pm0.13}$ | 77.1$_{\pm3.65}$ | 90.9$_{\pm0.31}$ | 85.6 |
| LAVA (ours) | 0.3M | **87.3**$_{\pm0.10}$ | 94.0$_{\pm0.51}$ | **90.1**$_{\pm2.67}$ | 63.3$_{\pm2.36}$ | **93.5**$_{\pm0.97}$ | 90.6$_{\pm0.07}$ | 77.4$_{\pm2.67}$ | **91.0**$_{\pm0.21}$ | **85.9** |

**Main results.** Table 2 shows experiment results on the GLUE benchmark: our LAVA method outperforms all the baseline methods and achieves the highest average score (85.9%), **0.3%** higher than DeLoRA and **1.2%** higher than LoRA. Additionally, for MRPC, CoLA, QNLI, and STS-B, our method consistently outperforms FT. For MRPC, our method even achieves superior accuracy over FT by **2.2%**. Compared with other PEFT methods, LAVA beats all other methods except on CoLA and RTE. But on these two datasets, performances are heavily influenced by random seeds, and we argue that the under-performances are not representative. The experimental results suggest that the integration of orthogonal regularization could help explore the low-rank spaces better and enhance learning capability.

## 5.2 COMMONSENSE REASONING

Then we scale the size of the model up to Gemma2-2b and even to LLaMA2-7b (Touvron et al., 2023) and turn to commonsense reasoning tasks to validate LAVA's compatibility in large models at a larger scale.

In this experiment, we evaluate LAVA against LoRA and include ChatGPT's accuracy obtained with gpt-3.5-turbo API using a zero-shot Chain of Thought (Wei et al., 2022). For fair comparison, we set the rank of all the finetuning methods as 32.

From the results in Table 3, we observe that orthogonal regularization in LAVA significantly improves LoRA's learning ability and enhances LoRA's performance, achieving +1.9% in LLaMA2-7b, even far out-performing the closed-resource model GPT-3.5-turbo. This shows that our LAVA

method can be scaled to large models as well, demonstrating its generality across LLMs with different sizes. Detailed results on all the datasets are shown in Appendix A.6.

Next, we evaluate LAVA's versatility and switch to models in vision and multimodal because these models normally have convolution to extract regional information and attention for global modeling.

## 5.3 Semantic Segmentation

We finetune SAM (Kirillov et al., 2023) using LAVA in three real-world scenarios, medical (Polyp (Bernal et al., 2015a) (Bernal et al., 2015b) and ISIC2017 (Codella et al., 2018)), natural (camouflaged object detection (Fan et al., 2020) and shadow (Vicente et al., 2016)) and agricultural (leaf disease segmentation (Rath, 2023)) respectively.

Table 3: Performance comparisons on Gemma2-2b and LLaMA2-7b on eight commonsense reasoning datasets. The best result is marked **bold**.

| Model | Method | # Params (%) | Avg. |
|---|---|---|---|
| ChatGPT | - | - | 77.0 |
| Gemma2-2b | LoRA | 1.07 | 77.4 |
| | VeRA | 0.02 | 70.2 |
| | LoRA+ | 1.07 | 77.5 |
| | DoRA | 1.09 | 77.3 |
| | LAVA | 1.07 | **78.1** |
| LLaMA2-7b | LoRA | 0.83 | 77.6 |
| | LAVA | 0.83 | **79.5** |

Table 4: Comparisons of different methods finetuning Depth-Anything. The best result is marked **bold**. Encoder-only means we only tune the encoder part of the model. † means the value is taken from (Yang et al., 2024).

| Method | params | $\delta_1 \uparrow$ | $\delta_2 \uparrow$ | $\delta_3 \uparrow$ | AbsRel $\downarrow$ | RMSE $\downarrow$ | log10 $\downarrow$ |
|---|---|---|---|---|---|---|---|
| FT (ZoeDepth)† | - | 0.951 | 0.994 | 0.999 | 0.077 | 0.282 | 0.033 |
| FT (Depth-Anything) | 335.3M | 0.984 | 0.998 | 1.000 | 0.056 | 0.206 | 0.024 |
| LoRA (Encoder-only) | 0.79M | 0.941 | 0.992 | 0.998 | 0.079 | 0.346 | 0.108 |
| LoRA | 2.04M | 0.937 | 0.994 | 0.999 | 0.090 | 0.356 | 0.039 |
| SVDiff | 0.11M | 0.916 | 0.991 | 0.998 | 0.102 | 0.399 | 0.044 |
| VeRA | 0.57M | 0.969 | 0.997 | 0.999 | 0.728 | 0.286 | 0.032 |
| Conv-Adapter | 1.21M | 0.969 | 0.997 | 0.999 | 0.072 | 0.278 | 0.031 |
| DoRA | 2.14M | 0.967 | 0.997 | 0.999 | 0.074 | 0.286 | 0.032 |
| LAVA | 1.14M | **0.972** | **0.997** | **0.999** | **0.070** | **0.274** | **0.030** |

We evaluate our method against LoRA (Hu et al., 2022) and Conv-LoRA (Zhong et al., 2024) on all three fields. All the settings follow the experiment configurations from Zhong et al. (2024). More details of datasets, baselines, and hyperparameters can be found in Appendix A.7.

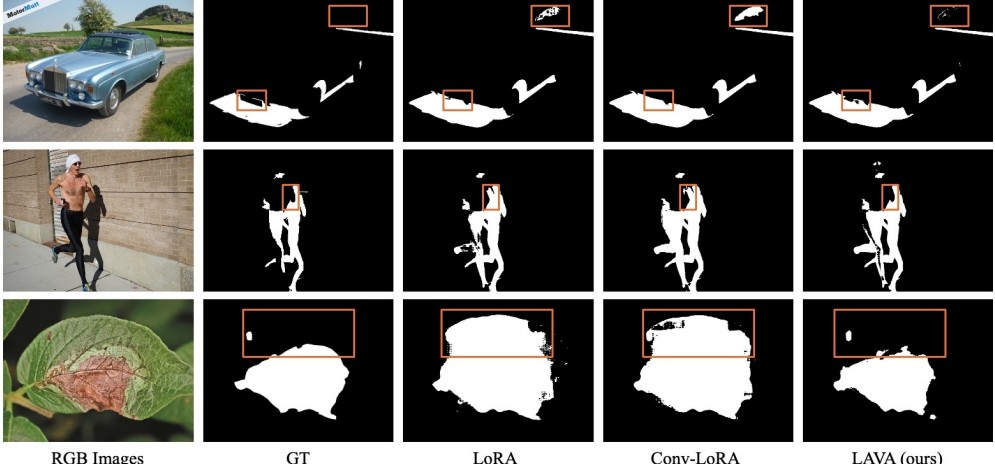

Figure 3: Comparisons of LoRA, Conv-LoRA, and LAVA (ours) in binary-class shadow segmentation (**top 2 rows**) and binary-class leaf disease segmentation (**bottom**).

Fig. 3 shows that LAVA can reinforce image-related local priors, which helps SAM to separate target regions from the background, while LoRA and Conv-LoRA still demonstrate difficulty in separating shadows from black objects. For example, in the second row, these two methods mistakenly recognize the man's arm as the shadow. Quantitative comparisons are shown in Table 13 in Appendix A.7. Here we keep the metrics the same as Conv-LoRA (Zhong et al., 2024).

## 5.4 DEPTH ESTIMATION

Similarly, we scale the proportion of convolution up, and change to depth estimation, where DPT (Ranftl et al., 2021) part (decoder part) of the pre-trained model is made up of convolution blocks. Specifically, in this experiment, we fine-tune Depth-Anything on nyu-depth v2 (Nathan Silberman & Fergus, 2012). All the experimental details are kept the same with Yang et al. (2024). Performances are evaluated using $\delta_1$, $\delta_2$, $\delta_3$, AbsRel, RMSE and log10. More details regarding the baselines are given in Appendix A.8.

**Quantitative comparison.** The results in Table 4 show that our proposed method yields much stronger and accurate depth estimation compared with other finetuning methods. We observe that our method is superior to ZoeDepth, which can be thought of as a previous SOTA method using significantly fewer parameters. Notably, for metric $\delta_1$, our method achieves 0.972, which is 2.1% higher than ZoeDepth. Although there exists a small margin (0.012) with full finetuning Depth Anything, considering the fact that our method only tunes 1.5% of total parameters, such a gap could be accepted and our method still shows potential. It is worth noting that the accuracy of LoRA is less than that of Encoder-only LoRA, meaning that directly using reshape-involved LoRA on convolution layers will degrade the overall performance, which further validates the importance of reshape-free finetuning.

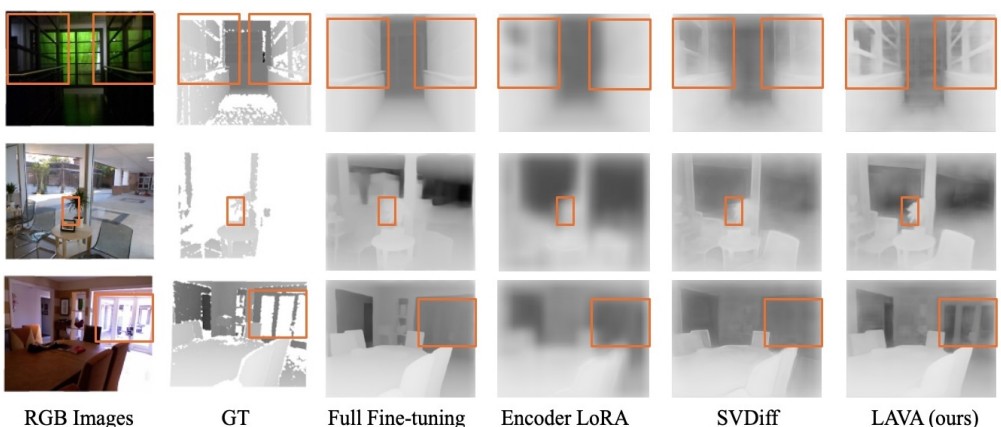

RGB Images    GT    Full Fine-tuning    Encoder LoRA    SVDiff    LAVA (ours)

Figure 4: Demonstrations of different finetuning methods on Depth-Anything

**Qualitative comparison.** Additionally, we provide qualitative comparisons in Fig. 4. Although our method underperforms full-finetuning in metrics, it demonstrates strong robustness in challenging scenarios: From the first and third row, we can observe that LAVA generalizes well in situations with mirrors, which we believe is challenging as the reflection of mirrors will become harder to predict the real depth information. From the second row, we can find that LAVA is robust in scenes with drastic changes in light intensity. These regions are highlighted in orange boxes. Further qualitative results are shown in Figs. 8 and 9 in Appendix A.8.

## 5.5 TEXT-TO-IMAGE GENERATION

Finally, we consider using LAVA in finetuning SDXL (Podell et al., 2023), utilizing the training scripts developed by HuggingFace. The target datasets, 3D icons, consist of 23 images, each one showing a company icon in a 3D version with a round-edge square at behind. The random seeds are kept the same for LoRA and our method for fair comparison. We fix the learning rate at 5e-4 and fine-tune the model for 10 epochs following the text-to-image pipeline[1]. Other hyperparameters are set to default values pre-defined in the script. For quantitative comparisons, we use metric FID (Heusel et al., 2017) to compare the similarity of generated images to real ones. The generated images are shown in Fig. 5 and quantitative results are presented in Appendix A.9. More generated images can be found in Fig. 10 in Appendix.

---

[1]Refer to https://github.com/huggingface/diffusers/tree/main/examples/text_to_image

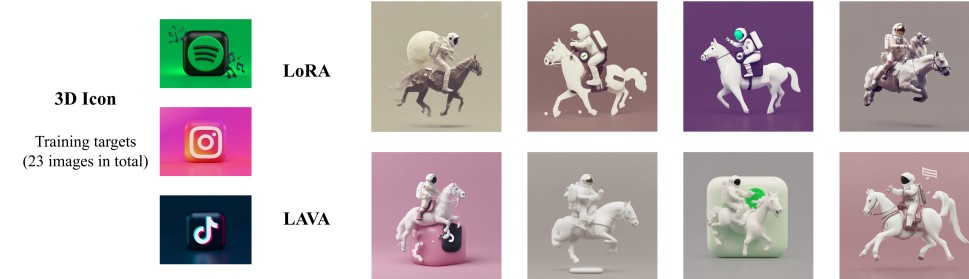

Prompt: a Spotify icon of an astronaut riding a horse, in the style of Spotify.

Figure 5: Images generated by SDXL with different finetuning methods on the 3d Icon dataset.

This result indicates that our method achieves better tuning performances compared with LoRA. For example, in 3d icon datasets, we can observe that for our method, they have clean lines and no redundant details. What's more, they use less saturated colors, which are closer to the Spotify icon. In contrast, images generated by LoRA have sophisticated lines (Column 1 and 4) and unnatural reflection effect (Column 4), meaning that LAVA could better understand the meaning of the style.

## 6 IN-DEPTH ANALYSIS

Compared with LoRA, our method has an additional hyperparameter: $\lambda$. In LAVA, rank $R$ controls what's the dimension of the subspace, and $\lambda$ controls the convergence rate towards a column-orthogonal matrix. In this section, we analyze different choices of $\lambda$ and rank $R$, and we hope it will give insights into how to tune these two parameters in downstream tasks.

**Robustness of $\lambda$**: We choose $\lambda = 0.1$ as the base, and then test the performances under different $\lambda$ settings on NLU and CIFAR10 classification. For example, the value 2 indicates twice the base, or $\lambda = 0.2$; other values are chosen similarly.

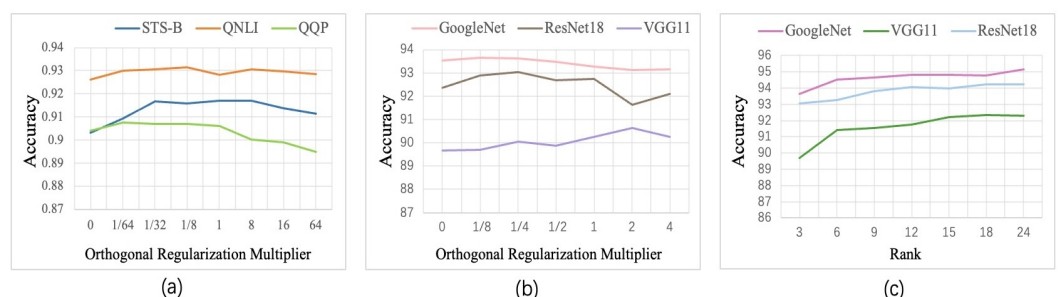

Figure 6: Hyperparameter analysis for LAVA. (a): Performances on NLU with different $\lambda$ settings. (b): Classification performances on CIFAR10 with different $\lambda$ settings on three model architectures. (c): Classification performances on CIFAR10 with different rank $R$ settings on three model architectures.

As shown in Figs. 6(a) and 6(b), our analysis shows that LAVA is able to achieve robustness on tuning both attention and convolution across a wide range of $\lambda$ values: ranging from $1/64$ of $\lambda = 0.1$ to $64\times$ of $\lambda = 0.1$. Such a phenomenon could guide us on how to tune $\lambda$ in other downstream tasks: we suggest in NLP tasks, choose a relatively small $\lambda$ value (from $1/32\times \lambda = 0.1$ to $\lambda = 0.1$; in vision tasks, the range from $1/2\times$ to $2\times$ of base $\lambda$ is more preferable.

**Applicable regions of rank $R$**: We further investigate the behaviors of ranks from 3 to 24 and fix orthogonal regularization term to be at $1\times$ of $\lambda = 0.1$ and test the relevant performances in finetuning CIFAR10 datasets to explore the learning ability of LAVA. This provides us with a quantitative measure of the joint impact of both $\lambda$ and rank $R$. In Fig. 6(c), we show that the classification precisions increase slightly as rank increases. This meets our expectation: each tensor-1 component in LAVA can be recognized as learning from the residual, and the main contribution can be learned

from the first few dimensions (there is a performance leap around points at rank 6). Surprisingly, we also observe that, after the main contribution is learned at rank 6, increasing ranks further can still enhance the performance. This means that our LAVA could help alleviate the noise introduced by increasing ranks. We attribute this to our orthogonal regularization, which stabilizes the training and ensures the full exploration in low-rank subspaces.

**How does orthogonal regularization affects the training dynamics?** As discussed in Section 4.2, adding orthogonal regularization can help stabilize the variance of the gradient during training. In this section, we provide experimental results to give intuitive empirical evidence. We compare the variances of gradients on each layer of the model with and without orthogonal regularization. As shown in Fig. 7, when training RoBERTa-base on the QNLI dataset, orthogonal regularization achieves significantly lower gradient variance, which is beneficial to reduce the effect of noise and avoid poor local solutions. More details of the gradient variances of different layers on additional datasets can be found in Fig. 12 in the Appendix.

**What is the additional computational overhead for orthogonal regularization?**

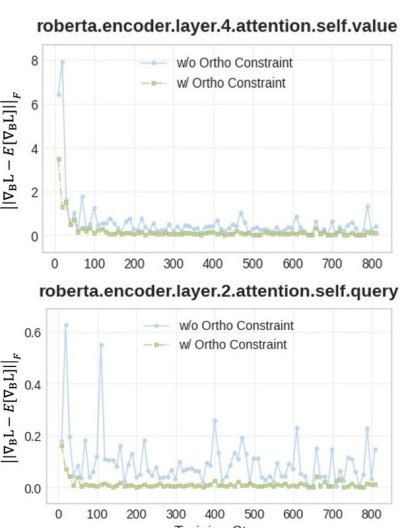

Figure 7: Variance analysis of the component $B$ during training.

We conduct the computational efficiency analysis on the additional cost of orthogonal regularization in training. We provide theoretical computation analysis and memory usage in Table 15 in the Appendix. Here, we consider the input as a sequence of tokens, and FF stands for feedforward pass, BW: backward pass, LC: loss computation, TP: trainable parameters, CC: computational cost including forward, backward pass, and loss calculation, $T$ stands for the cost of calculating next-token prediction loss, which is widely used in finetuning. We follow the setting $m = n$ here to simplify the analysis. Assuming input $x \in \mathbb{R}^{S \times n}$, $S$ is the sequence length, $n$ is the hidden size of the model, and $r$ is the rank pre-defined before training. Since $r \ll S$ in the long-sequence setting, the additional cost $3rn^2$ is negligible in the training process.

Besides theoretical analysis, we also compare the wall-clock time spent for forward, backward, and loss calculation phases of 1 epoch. Here we use the time of LoRA as the base, and compare how many times longer the time cost is compared to other methods and LoRA.

As shown in Table 16 in the Appendix, compared with vanilla LoRA, the overhead of orthogonal regularization can be maintained at not more than 20%. Besides, the overall computation of DoRA is far more than that of vanilla LoRA, making it less applicable in the resource-constrained finetuning phase. In a convolution situation, the overhead of LAVA is still within 20% budget threshold, indicating that LAVA achieves a balance between accuracy and efficiency.

## 7 CONCLUSION

In this work, we first conducted a novel increment analysis experiment to empirically validate the existence of intrinsic dimensions in convolution layers, which is similar to that in LoRA. Inspired by the observations from the experiment, we proposed LAVA to generalize the update of the matrix and convolution together. Moreover, we analyzed the training dynamics and presented to use orthogonal regularization in this new parameterization form. We also provided the theoretical analysis that it can reduce the variance of trainable parameters. LAVA consistently outperforms LoRA for various downstreaming tasks and model architectures. However, we are also aware that we fix the learning rates of matrices and convolution layers the same during training, making the optimization process sub-optimal. For future work, we wish to make innovations to make the learning rate adaptive towards different modules and apply our method to video generation fields and other modalities.

## REPRODUCIBILITY STATEMENT

We provide the source codes anonymously in the given link: `https://anonymous.4open.science/r/test-ABC7946/`. Mathematical proofs are provided in Appendix A.2 and A.3.

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

# A APPENDIX

## A.1 THE USE OF LARGE LANGUAGE MODELS (LLMS)

LLMs are used to polish up the writing in the abstract and introduction parts.

## A.2 PROOF OF THEOREM 1

**Theorem 1**: If the optimizer is SGD, denote the full gradient as $\bar{G}(t)$, gradient calculated on a mini-batch as $G(t)$, $G(t)$ satiesfies: $\mathbb{E}[G(t)] = \bar{G}(t)$, $\mathbb{E}[||G(t) - \bar{G}(t)||_2^2] \leq \sigma^2$. Assuming $||\bar{G}(t)||_2 \leq \mu$, when the regularization strength term is set to $\lambda$, $A$ will get converged to orthogonal matrix $M$ ($M^\top M = I$) at the speed of $\mathcal{O}(\frac{\mu+\sigma}{\lambda})$.

**Proof.**

If the optimizer is SGD, denote $D_k = A_k^\top A_k - I$, $G_k = \nabla_W L(A_k, B_k)$, $C_k = A_k^\top A_k$, then the parameter update has the following form:

$$\begin{cases} A_{k+1} \leftarrow A_k - \eta[B_k^\top G_k + 4\lambda A_k(A_k^\top A_k - I)] = A_k(I - 4\eta\lambda D_k) - \eta B_k^\top G_k, \\ B_{k+1} \leftarrow B_k - \eta G_k A_k^\top. \end{cases} \quad (6)$$

Then

$$\begin{aligned} C_{k+1} &= A_{k+1}^\top A_{k+1} \\ &= [(I - 4\eta\lambda D_k)^\top A_k^\top - \eta G_k^\top B_k][A_k(I - 4\eta\lambda D_k) - \eta B_k^\top G_k] \\ &= (I - 4\lambda\eta D_k)^\top C_k(I - 4\lambda\eta D_k) - \\ &\quad \eta(1 - 4\lambda\eta D_k)^\top A_k^\top B_k^\top G_k - \eta G_k^\top B_k A_k(I - 4\lambda\eta D_k) + \mathcal{O}(\eta^2 I_r) \\ &= C_k - 4\lambda\eta C_k D_k - 4\lambda\eta D_k^\top C_k - \eta(A_k^\top B_k^\top G_k + G_k^\top B_k A_k) + \mathcal{O}(\eta^2 I_r) \\ &= C_k - 8\eta\lambda C_k(C_k - I) - \eta(A_k^\top B_k^\top G_k + G_k^\top B_k A_k) + \mathcal{O}(\eta^2 I_r) \\ &\approx C_k - 8\eta\lambda C_k(C_k - I) - \eta(A_k^\top B_k^\top G_k + G_k^\top B_k A_k). \end{aligned} \quad (7)$$

When training LoRA, the learning rate $\eta$ is often set to be at the scale of 1e-4, the discrete optimization process can be interpreted as a numerical integration of a continuous-time dynamical system, or **gradient flow**. Denote $H = A^\top B^\top G + G^\top B A$, then we get the following formulas:

$$\begin{cases} \frac{dA}{dt} = -B^\top G - 4\lambda A(A^\top A - I), \\ \frac{dB}{dt} = -GA^\top, \\ \frac{dC}{dt} = -8\lambda C(C - I) - H, \\ \frac{dD}{dt} = -8\lambda(D^2 + D) - H. \end{cases} \quad (8)$$

Let $S(t) := ||A^\top A - I||_F^2 = ||D||_F^2 = tr(D^2)$,

$$\begin{aligned} \frac{dS(t)}{dt} &= tr\left(D\frac{dD}{dt} + \frac{dD}{dt}D\right) \\ &= tr\left(-8\lambda D(D^2 + D) - DH - 8\lambda(D^2 + D)D - HD\right) \\ &= -2tr(DH) - 16\lambda tr(D(D^2 + D)). \end{aligned} \quad (9)$$

Now we give the upper bound of $\frac{dS(t)}{dt}$:

$$\begin{aligned} |tr(DH)| &= <D, H>_F \leq ||D||_F ||H||_F \\ &\leq 2||D||_F \cdot ||B^\top G A^\top||_F \\ &\leq 2||D||_F \cdot ||B||_F \cdot ||G||_2 \cdot ||A||_F. \end{aligned} \quad (10)$$

Since $tr(D(D^2 + D)) = tr(D^3) + tr(D^2) \geq ||D||_F^2 - ||D||_F^3$, then

$$S'(t) \leq 4||D||_F ||B||_F ||G||_2 ||A||_F - 16\lambda(||D||_F^2 - ||D||_F^3). \tag{11}$$

Before continuing, we give two lemmas to help further proof.

**Lemma 1: Right Derivative of Frobenius Norm**. Let $A(t)$ be a matrix-valued function that is differentiable at $t$, and define $a(t) = ||A(t)||_F$. Then the right derivative of $a(t)$ satisfies:

$$a'_+(t) \leq ||A'(t)||_F$$

**Proof.**

Let $a(t) = ||A(t)||_F$, $g(t) = ||A(t)||_F^2 = a^2(t) = <A(t), A(t)>_F$,

then $g'(t) = 2a(t)a'(t) = 2 < A'(t), A(t) >_F \leq 2||A(t)||_F ||A'(t)||_F = 2a(t)||A'(t)||_F$.

If $a(t) > 0$, then $a'(t) \leq ||A'(t)||_F$, or $||A(t)||'_F \leq ||A'(t)||_F$, else $a(t) = 0$,

$$\begin{aligned}
a'_+(t) &= \lim_{h \to 0^+} \frac{||A(t+h)||_F - ||A(t)||_F}{h} \\
&= \lim_{h \to 0^+} \frac{||A(t+h)||_F}{h} \\
&= \lim_{h \to 0^+} ||\frac{1}{h}A(t+h)||_F \\
&= || \lim_{h \to 0^+} \frac{A(t+h) - A(t)}{h}||_F \\
&= ||A'(t)||_F. \tag{12}
\end{aligned}$$

We can get the conclusion that $[||A(t)||_F]'_+ \leq ||A'(t)||_F$.

**Lemma 2: Upper bound of Frobenius Norm** $||A(t)||_F$ and $||B(t)||_F$. Denote $\alpha = 2(G_{max} + 4\lambda\sqrt{m})$, $\beta = 8\lambda$. The norm of $||A(t)||_F$ and $||B(t)||_F$ have the following upper bound:

$$||A(t)||_F^2 + ||B(t)||_F^2 \leq \max\{||A_0||_F^2 + ||B_0||_F^2, \frac{\alpha}{\beta}\}. \tag{13}$$

**Proof.**

Based on (8), we get the following derivative:

$$A'(t) = -B(t)^\top G(t) - 4\lambda A(t)D(t), \quad B'(t) = -G(t)A(t)^\top. \tag{14}$$

Denote $G_{max} = \sup_{t \in [0,T]} ||G(t)||_2$, then $\mathbb{E}[G_{max}] \leq \mu + \sigma$.

Let $a(t) = ||A(t)||_F$, $b(t) = ||B(t)||_F$, then:

$$\begin{cases} ||A'(t)||_F \leq ||G||_2 b(t) + 4\lambda a(t)[a^2(t) + \sqrt{m}] \leq G_{max}b(t) + 4\lambda a(t)[a^2(t) + \sqrt{m}], \\ ||B'(t)||_F \leq G_{max}a(t). \end{cases} \tag{15}$$

Let $v(t) = a^2(t) + b^2(t)$, then $v'(t) = v'_+(t) = 2a'_+(t)a(t) + 2b'_+(t)b(t)$.

Based on Lemma 1, we can substitute $a'_+(t)$ and $b'_+(t)$ with $||A'(t)||_F$ and $||B'(t)||_F$:

$$\begin{aligned}
v'(t) &= 2a(t)a'(t) + 2b(t)b'(t) \\
&\leq 2a(t)(G_{max}b(t) + 4\lambda a(t)(a^2(t) + \sqrt{m})) + 2b(t)(G_{max}a(t)) \\
&\leq 4G_{max}a(t)b(t) + 8\lambda a^4(t) + 8\lambda a^2(t)\sqrt{m}. \tag{16}
\end{aligned}$$

Based on Young's inequality,

$$4G_{max}a(t)b(t) \leq 2G_{max}(a^2(t) + b^2(t)) = 2G_{max}v(t). \tag{17}$$

Then, we can get:

$$v'(t) \leq 2(G_{max} + 4\lambda\sqrt{m})v(t) + 8\lambda v^2(t). \tag{18}$$

Let $w'(t) = 2(G_{max} + 4\lambda\sqrt{m})w(t) + 8\lambda w^2(t)$, $w(0) = v(0)$.

We have the equation: $w'(t) = \alpha w(t) + \beta w^2(t)$, $\alpha = 2(G_{max} + 4\lambda\sqrt{m})$, $\beta = 8\lambda$.

Solve the above Bernoulli's equation, the final expression of $w(t)$ is:

$$w(t) = \frac{\alpha w_0 e^{\alpha t}}{\alpha + \beta w_0(e^{\alpha t} - 1)}, \tag{19}$$

where $\alpha = 2(G_{max} + 4\lambda\sqrt{m})$, $\beta = 8\lambda$.

Based on Comparison Principle, $v(t) \leq w(t)$, for $t \in [0, T]$. Thus,

$$v(t) \leq w(t) \leq \sup_{t \in [0,T]} w(t) = \begin{cases} v_0, \alpha \leq \beta v_0; \\ \frac{\alpha v_0}{(\alpha - \beta v_0)e^{-\alpha T} + \beta v_0}, \alpha > \beta v_0. \end{cases} \leq \max\{v_0, \frac{\alpha}{\beta}\} < +\infty, \tag{20}$$

where $v_0 = ||A_0||_F^2 + ||B_0||_F^2$, which concludes Lemma 2.

**Move back to (12),** denote $d(t) = ||D||_F$, $C(G_{max}, \lambda, m, ||A_0||_F^2 + ||B_0||_F^2, T) = \max\{v_0, \frac{\alpha}{\beta}\}$, then

$$2d(t)d'_+(t) \leq 4d(t)a(t)b(t)||G||_2 - 16\lambda(d^2(t) - d^3(t))$$
$$d'_+(t) \leq 2a(t)b(t)||G||_2 - 8\lambda(d(t) - d^2(t))$$
$$\leq -8\lambda d(t) + 8\lambda d^2(t) + 2C^2(G_{max}, \lambda, m, ||A_0||_F^2 + ||B_0||_F^2, T)G_{max}. \tag{21}$$

Let $f'(t) = -8\lambda f(t) + 8\lambda f^2(t) + 2C^2(G_{max}, \lambda, m, ||A_0||_F^2 + ||B_0||_F^2, T)G_{max}$, $f(0) = d(0) = ||A_0^\top A_0 - I||_F$.

We ignore the quadratic term on the right-hand side for the sake of simplicity. Based on Comparison Principle that $d(t) \leq f(t), \forall t \in [0, T]$, and $f(t) \approx ||A_0^\top A_0 - I||_F e^{-8\lambda t} + \frac{2C^2 G_{max}}{8\lambda}(1 - e^{-8\lambda t})$, we can get the following upper bound:

$$d(t) \approx ||A_0^\top A_0 - I||_F e^{-8\lambda t} + \frac{2C^2 G_{max}}{8\lambda}(1 - e^{-8\lambda t})$$
$$\leq ||A_0^\top A_0 - I||_F e^{-8\lambda t} + \frac{C^2(G_{max}, \lambda, m, ||A_0||_F^2 + ||B_0||_F^2, T)G_{max}}{4\lambda}. \tag{22}$$

Then the expectation has the following form:

$$\mathbb{E}[d(t)] \leq ||A_0^\top A_0 - I||_F e^{-8\lambda t} + \frac{C^2(\mu + \sigma, \lambda, m, ||A_0||_F^2 + ||B_0||_F^2, T)(\mu + \sigma)}{4\lambda}$$
$$\sim \mathcal{O}(\frac{\max\{||A_0||_F^2 + ||B_0||_F^2, \frac{\alpha}{\beta}\}(\mu + \sigma)}{4\lambda})$$
$$= \mathcal{O}(\frac{\max\{||A_0||_F^2 + ||B_0||_F^2, \frac{\mu+\sigma}{4\lambda} + \frac{\sqrt{m}}{2}\}(\mu + \sigma)}{\lambda})$$
$$\sim \mathcal{O}(\frac{\mu + \sigma}{\lambda}). \tag{23}$$

### A.3 Proof of proposition: Upper bound of gradient variance

**Proof.**

Let $G = \nabla_W L \in \mathbb{R}^{m \times n}$, $\Sigma_G = \mathbb{E}[(G - \mathbb{E}G)^\top (G - \mathbb{E}G)] \in \mathbb{R}^{n \times n}$. Based on the definition of variance:

$$
\begin{aligned}
\text{Var}(\nabla_B L) &= \mathbb{E}||GA^\top - \mathbb{E}[GA^\top]||_F^2 \\
&= \mathbb{E}[tr((GA^\top - \mathbb{E}[GA^\top])^\top (GA^\top - \mathbb{E}[GA^\top]))] \\
&= tr(\mathbb{E}[(GA^\top - \mathbb{E}[GA^\top])^\top (GA^\top - \mathbb{E}[GA^\top])]) \\
&= tr(A\mathbb{E}[(G - \mathbb{E}G)^\top (G - \mathbb{E}G)]A^\top) \\
&= tr(A\Sigma_G A^\top) \\
&= tr(A^\top A \Sigma_G) \leq tr(||A^\top A||_2 \Sigma_G) \quad (*) \\
&\leq (1 + E_A)tr(\Sigma_G).
\end{aligned}
\tag{24}
$$

Next we prove $(*)$: if $B \succeq 0$, then $tr(AB) \leq ||A||_2 tr(B) G_{max}$ .

Since $B$ is positive semi-definite, $B$ can be diagonalized into the following form: $B = Q\Lambda Q^\top$. Denote matrix $D = Q^\top A Q$, then,

$$
\begin{aligned}
tr(AB) &= tr(AQ\Lambda Q^\top) = tr(Q^\top A Q \Lambda) \\
&= tr(D\Lambda) = \sum_i d_{ii} \cdot \lambda_i \\
&\leq \sum_i ||D||_2 \lambda_i \\
&= \sum_i ||A||_2 \cdot \lambda_i \\
&= ||A||_2 tr(B).
\end{aligned}
\tag{25}
$$

### A.4 Toy experiment

We finetune ResNet18, ResNet34, VGG11, AlexNet, GoogLeNet and ConvNeXt on CIFAR10 using the optimizer AdamW (Loshchilov & Hutter, 2017). The learning rate is set to be 1e-3, betas for the optimizer are (0.9, 0.999), and weight decay is set to be 0. In this experiment, to analyze the low-rank subspaces, we don't set the orthogonal regularization term. Table 5 lists the results of three methods in all the models.

Table 5: Comparisons of reshape-involved, reshape-free and full-finetuning methods on CIFAR10.

| Model | Metric | AlexNet | VGG11 | ResNet34 | GoogLeNet | ResNet18 | ConvNeXt |
|---|---|---|---|---|---|---|---|
| **Reshape-involved** | Trainable parameters(%) | 0.13 | 0.09 | 1.05 | 2.32 | 1.01 | 3.12 |
| | Accuracy | 74.21 | 60.47 | 78.58 | 79.44 | 76.16 | 76.09 |
| **Reshape-free** | Trainable parameters(%) | 0.08 | 0.04 | 0.25 | 1.66 | 0.28 | 0.15 |
| | Accuracy | 87.27 | 90.09 | 93.85 | 93.52 | 92.37 | 98 |
| **FT** | Trainable parameters(%) | - | - | - | - | - | - |
| | Accuracy | 86.81 | 90.94 | 92.63 | 93.83 | 92.88 | 95.34 |

## A.5 NLU EXPERIMENT DETAILS

The GLUE benchmark (General Language Understanding Evaluation) is a widely used collection of datasets and evaluation metrics designed to assess the performance of natural language understanding (NLU) models. Following Wu et al. (2024a), we split the publicly available validation dataset into two parts: if the validation dataset is larger than 2K, then 1K is chosen as a new validation set; otherwise, half of it is chosen as new validation set. Table 6 shows the detailed split and metric used for each dataset in GLUE benchmark. Here, MCC represents the Matthews correlation coefficient, ACC represents accuracy, and CORR represents Pearson correlation coefficient.

Table 6: Splits and metrics of the GLUE benchmark.

| Split Sizes | MNLI | SST-2 | MRPC | CoLA | QNLI | QQP | RTE | STS-B |
|---|---|---|---|---|---|---|---|---|
| **#Train** | 393K | 67K | 3.7K | 8.5K | 105K | 364K | 2.5K | 5.7K |
| **#Validation** | 1K | 436 | 204 | 522 | 1K | 1K | 139 | 750 |
| **#Test** | 8K | 436 | 204 | 521 | 4.5K | 39K | 138 | 750 |
| **Metric** | ACC | ACC | ACC | MCC | ACC | ACC | ACC | CORR |

We provide details of the chosen baselines below:

- *Full Finetune (FT)*: Full fine-tuning is a common approach for adaptation. During adaptation, all parameters of the model undergo gradient updates.
- *Adapter*: Adapter (Houlsby et al., 2019) inserts lightweight and trainable modules between two sub-layers of the transformer.
- *Adapter-FFN*: Adapter-FFN (Pfeiffer et al., 2020) is a variant of Adapter method, it inserts lightweight and trainable modules after FFN sub-layer in transformer architecture.
- *BitFit*: Bias-only (BitFit) is an early work proposed by Ben Zaken et al. (2022): it only tuns bias-terms of the model.
- *LoReFT*: Low-rank Linear Subspace ReFT, or LoReFT (Wu et al., 2024b), learns task-specific interventions on hidden representations to adapt the pre-trained model on down-streaming tasks.
- *RED*: Representation Editing or (RED) (Wu et al., 2024a), similar to LoReFT, is a method to modify representations generated at some layers using scaling and biasing operations.
- *LoRA*: Vanilla LoRA, trains two matrices, one for down-projection the input into low-rank spaces, and one for up-projecting it back.
- *DeLoRA*: Bini et al. (2025) decouples the weight into two components, direction and normalization. By bounding the distance of the transformation, it can enhance the robustness.

For all the experiments, the maximum sequence length is set to be 512. Other hyperparameters are chosen by grid search. All the experiment results are averaged on random seeds 42, 43, 44 45 and 46. Hyperparameters are reported in Table 7.

Table 7: Hyperparameters for LAVA in NLU

| Hyperparameters | MNLI | SST-2 | MRPC | CoLA | QNLI | QQP | RTE | STS-B |
|---|---|---|---|---|---|---|---|---|
| Rank | | | | 8 | | | | |
| Dropout | | | | 0.1 | | | | |
| alpha | | | | 8 | | | | |
| Max Seq | | | | 512 | | | | |
| Learning Rate $\eta$ | 5e-4 | 5e-4 | 3e-4 | 4e-4 | 4e-4 | 5e-4 | 5e-4 | 4e-4 |
| Epoch | 30 | 20 | 20 | 80 | 25 | 25 | 80 | 40 |
| Batch size | 16 | 8 | 4 | 32 | 32 | 64 | 32 | 32 |
| orthogonal regularization | 0.3 | 1.0 | 1.0 | 0.05 | 0.025 | 0.1 | 0.1 | 0.05 |

### A.6 Commonsense Reasoning Experiment

For fair comparison, we set the rank to be 32, which is the same as experiments in Hu et al. (2023), and we only tune the learning rate and orthogonal regularization term $\lambda$.

Table 8: Performance comparisons on Gemma2-2b, LLaMA3-1b and LLaMA2-7b on eight commonsense reasoning datasets. Best result is marked **bold**.

| Model | Method | # Params (%) | BoolQ | PIQA | SIQA | HellaSwag | WinoGrande | ARC-e | ARC-c | OBQA | Avg. |
|---|---|---|---|---|---|---|---|---|---|---|---|
| ChatGPT | - | - | 73.1 | 85.4 | 68.5 | 78.5 | 66.1 | 89.8 | 79.9 | 74.8 | 77.0 |
| Gemma2-2b | LoRA | 1.07 | 68.5 | 80.5 | 77.2 | 86.9 | 78.5 | 81.8 | 66.0 | **79.6** | 77.4 |
| | VeRA | 0.02 | 65.6 | 75.4 | 74.6 | 59.3 | 72.8 | 80.0 | 63.0 | 70.6 | 70.1 |
| | LoRA+ | 1.07 | **68.6** | **81.1** | **77.7** | **89.4** | 77.6 | **84.1** | **66.8** | 75.0 | 77.5 |
| | DoRA | 1.09 | 67.5 | 80.7 | **77.7** | 87.2 | 79.2 | 81.6 | 66.7 | 77.8 | 77.3 |
| | LAVA | 1.07 | 68.5 | 80.8 | 77.6 | 88.8 | **80.4** | 83.2 | 66.6 | 78.6 | **78.2** |
| LLaMA2-7b | LoRA | 0.83 | 69.8 | 79.9 | **79.5** | 83.6 | **82.6** | 79.8 | 64.7 | **81.0** | 77.6 |
| | LAVA | 0.83 | **71.9** | **84.2** | 78.3 | **86.6** | 81.9 | **83.4** | **68.6** | 80.8 | **79.5** |

Table 8 shows all the results on eight datasets for our method. LAVA almost outperforms the performances of all the baselines on every dataset.

### A.7 Semantic Segmentation Experiment Details

#### A.7.1 Hyperparameters for the experiment

In semantic segmentation experiments, we follow the setting of Conv-LoRA (Zhong et al., 2024) and test our method on datasets about polyp, skin lesion, camouflaged object, leaf disease and shadow segmentation. Table 9 lists out the mapping between training and testing datasets. For detailed descriptions and dataset configurations, please refer to Zhong et al. (2024).

In this experiment, we compare LAVA against two baselines: LoRA and Conv-LoRA. Conv-LoRA introduces MoE structure (Shazeer et al., 2017) into LoRA. On each path of MoE, Conv-LoRA injects lightweight and trainable convolution layers to extract features at different scale. Such a method could introduce image-related inductive bias at different scales, which benefits the performances.

Table 9: Details of train and testing datasets

| **Train** | Polyp | ISIC 2017 | CAMO | SBU | Leaf |
|---|---|---|---|---|---|
| **Test** | CVC-612 | ISIC 2017 | CAMO | SBU | Leaf |

Tables 10, 11 and 12 provide hyperparameters used in this experiment.

Table 10: Hyperparameters for Conv-LoRA used in semantic segmentation

| **Dataset** | **Rank** | **# experts** | **Batch size** | **Learning Rate** $\eta$ | **Epoch** | **Metric** | **Loss** |
|---|---|---|---|---|---|---|---|
| Polyp | 3 | 8 | 4 | 1e-4 | 30 | sm | structure_loss |
| CAMO | 3 | 8 | 4 | 1e-4 | 20 | sm | structure_loss |
| Leaf | 3 | 8 | 4 | 3e-4 | 30 | iou | structure_loss |
| ISIC2017 | 3 | 8 | 4 | 1e-3 | 30 | iou | structure_loss |
| SBU | 3 | 8 | 4 | 1e-4 | 10 | ber | balanced_bce |

Table 11: Hyperparameters for LoRA used in semantic segmentation

| Dataset | Rank | Alpha | Batch size | Learning Rate $\eta$ | Epoch | Metric | Loss |
|---------|------|-------|------------|----------------------|-------|--------|------|
| Polyp | 3 | 32 | 4 | 1e-4 | 30 | sm | structure_loss |
| CAMO | 3 | 32 | 4 | 1e-4 | 20 | sm | structure_loss |
| Leaf | 3 | 32 | 4 | 3e-4 | 30 | iou | structure_loss |
| ISIC2017 | 32 | 3 | 4 | 1e-3 | 30 | iou | structure_loss |
| SBU | 3 | 32 | 4 | 1e-4 | 10 | ber | balanced_bce |

Table 12: Hyperparameters for LAVA used in semantic segmentation

| Dataset | Rank | Alpha | Orthogonal strength Multiplier $\lambda$ | Batch size | Learning Rate $\eta$ | Epoch | Metric | Loss |
|---------|------|-------|------------------------------------------|------------|----------------------|-------|--------|------|
| Polyp | 3 | 32 | 0.5 | 4 | 1e-4 | 30 | sm | structure_loss |
| CAMO | 3 | 32 | 0.1 | 4 | 1e-4 | 20 | sm | structure_loss |
| Leaf | 3 | 32 | 0.1 | 4 | 3e-4 | 30 | iou | structure_loss |
| ISIC2017 | 3 | 32 | 1.0 | 4 | 1e-3 | 30 | iou | structure_loss |
| SBU | 3 | 32 | 0.1 | 4 | 1e-4 | 10 | ber | balanced_bce |

### A.7.2 QUANTITATIVE COMPARISONS IN SEMANTIC SEGMENTATION EXPERIMENT

Table 13: Performance comparison across different domains. Best result is marked **bold**. The second highest value is marked using underline. All the experiment results are averaged on random seeds 42, 43 and 44.

| Method | Ratio | Medical | | | | Natural Images | | | | Agriculture | |
|--------|-------|---------|---|---|---|----------------|---|---|---|-------------|---|
| | | CVC-612 | | ISIC 2017 | | CAMO | | | SBU | Leaf | |
| | | $S_\alpha$ ↑ | $E_\phi$ ↑ | Jac ↑ | Dice ↑ | $S_\alpha$ ↑ | $E_\phi$ ↑ | $F_\beta$ ↑ | BER ↓ | IoU ↑ | Dice ↑ |
| LoRA | 4.00 / 0.62% | $90.3_{\pm0.63}$ | $91.8_{\pm0.92}$ | $\underline{77.3}_{\pm0.66}$ | $\underline{87.2}_{\pm0.42}$ | $\mathbf{88.6}_{\pm0.52}$ | $\underline{92.4}_{\pm0.65}$ | $83.5_{\pm1.13}$ | $2.86_{\pm0.11}$ | $72.2_{\pm0.84}$ | $83.9_{\pm0.56}$ |
| Conv-LoRA | 4.02 / 0.63% | $\underline{90.4}_{\pm0.62}$ | $\underline{92.1}_{\pm0.29}$ | $77.2_{\pm0.05}$ | $87.1_{\pm0.03}$ | $\underline{88.2}_{\pm0.25}$ | $91.8_{\pm0.16}$ | $\underline{83.6}_{\pm0.06}$ | $\underline{2.78}_{\pm0.03}$ | $\underline{73.1}_{\pm0.21}$ | $\underline{84.5}_{\pm0.14}$ |
| LAVA | 4.00 / 0.62% | $\mathbf{91.0}_{\pm0.73}$ | $\mathbf{93.4}_{\pm0.63}$ | $\mathbf{77.9}_{\pm0.41}$ | $\mathbf{87.6}_{\pm0.26}$ | $\mathbf{88.6}_{\pm0.08}$ | $\mathbf{92.6}_{\pm0.42}$ | $\mathbf{83.7}_{\pm0.42}$ | $\mathbf{2.64}_{\pm0.11}$ | $\mathbf{73.6}_{\pm0.25}$ | $\mathbf{84.8}_{\pm0.17}$ |

Table 13 provides the detailed quantitative results for all three methods. LAVA could improve the segmentation results further while keeping the trainable parameters at the same level compared with other two PEFT methods.

## A.8 DEPTH ESTIMATION EXPERIMENT DETAILS

In this experiment, we finetune Depth-Anything in metric depth estimation. Metric depth estimation is a field that predicts the absolute distance in real-world units from the camera to each pixel in an image. Firstly, we provide some descriptions of the chosen baselines (unmentioned baselines have been discussed in the main text):

- *SVDiff*: SVDiff can be recognized as a general framework to finetune both convolution and matrix. For convolution layers, it flattens the tensor $\mathcal{X} \in \mathbb{R}^{c_{out} \times c_{in} \times h \times w}$ into its matrix form $X \in \mathbb{R}^{c_{out} \times (c_{in} \times h \times w)}$, and then perform SVD and only tunes its singular values.
- *VeRA*: VeRA (Kopiczko et al., 2023) replaces learnable low-rank matrices (i.e., $A$ and $B$ in LoRA) with fixed random matrices and trains only two small scaling vectors $\lambda_d$ (per-rank) and $\lambda_b$ (per-output-dimension).

### A.8.1 HYPERPARAMETERS FOR DEPTH-ESTIMATION EXPERIMENT

To finetune convolution layers, LoRA reshapes the convolution weights into its matrix form and then tunes the matrix accordingly. For LoRA, we conduct two experiments: one to only tune the encoder part, and the other to tune both the encoder and decoder of the model. For SVDiff, we tune both components. We believe that this setting could prove the necessity to tune convolution parts in pixel-level granularity tasks. The batch size is set to be 4, and we run each baseline method for 10 epochs. For SVDiff, the learning rate is set to be 0.000161 at the beginning, and weight decay is set to be 0.01. For LoRA (both trained on encoder and encoder+decoder) and LAVA, the learning rate is set be 0.000161, and weight decay is the same with the setting in SVDiff. Additionally, the orthogonal regularization term of LAVA is set to be 0.01.

### A.8.2 MORE RESULTS TO COMPARE LAVA AGAINST FULL-FINETUNING

We provide more qualitative results in Fig. 8 to showcase LAVA's superiority in difficult situations compared with full fine-tuning. As discussed in the Qualitative Comparison parts before, LAVA could successfully recognize and calculate the real distance in regions with strong ambient light, perspective in transparent objects, and reflections. For example, in the first column of Fig. 8, the induction cooker on the table reflects sunlight. In this case, LAVA could still predict the item on the table, while full-finetuning fails to do so.

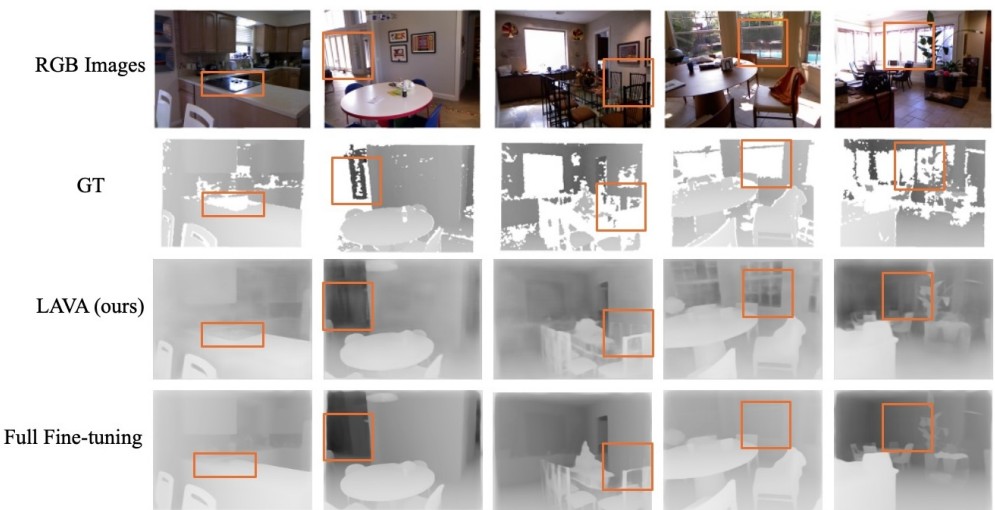

Figure 8: More qualitative results between LAVA and full-finetuning.

### A.8.3   MORE RESULTS TO COMPARE PEFT METHODS IN THE DEPTH-ESTIMATION TASK

In Fig. 9, it lists out the generated images from all the methods. Compared DoRA with LoRA, DoRA shows that weight normalization can help learn features better. However, the generated image is still relatively blurry, meaning that DoRA is not sufficient in processing reshape-involved tuning. Conv-Adapter is another competing baseline, but as shown in the fourth column, the outlines of the object have unwelcoming artifacts.

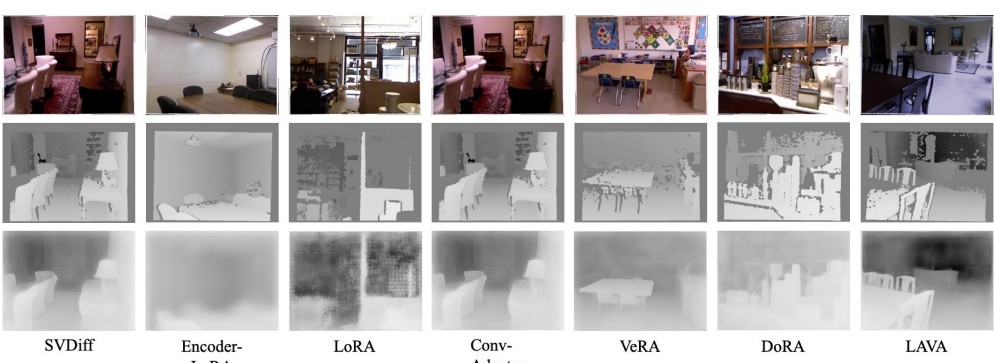

| SVDiff | Encoder-LoRA | LoRA | Conv-Adapter | VeRA | DoRA | LAVA |

Figure 9: More qualitative results between PEFT methods.

### A.9   TEXT-TO-IMAGE GENERATION EXPERIMENT DETAILS

FID (Heusel et al., 2017) and CLIP score are two widely used evaluation metrics for image generation models. Compared with metrics to compare pixels directly, they compare feature representations extracted from pre-trained Inception models. In Table 14, we provide FID and CLIP scores for LoRA and LAVA in five different random seed settings.

Table 14: FID comparisons

| Metric | LoRA | LAVA |
|---|---|---|
| **FID Score**↓ | 480.41 | 429.38 |
| **CLIP Score**↑ | 0.57 | 0.60 |

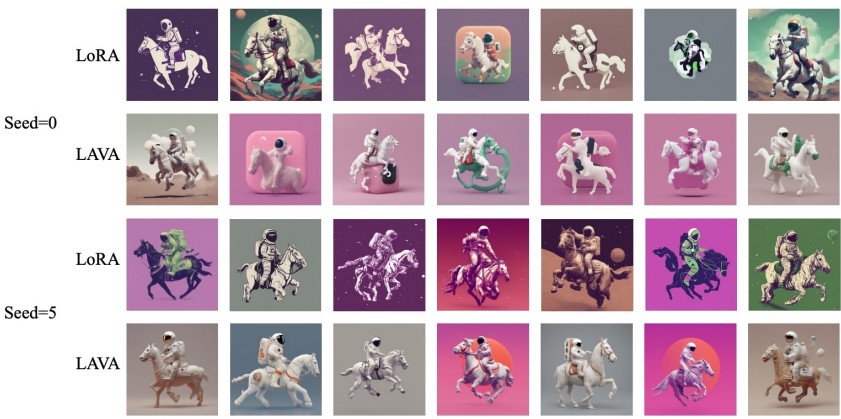

Figure 10: Comparisons of generated images from LoRA and LAVA.

## A.10 EFFECTS OF DIFFERENT REGULARIZATION METHODS ON PRE-TRAINED CONVOLUTIONAL NETWORKS.

For fair comparison, we fix the orthogonal regularization strength $\lambda$ at 0.025, and apply it on different blocks to compare the performances. As is shown in Fig. 11, we empirically find that imposing orthogonality on the channel-mode factors U and V (corresponding to output and input channels) yields the largest performance gains, while orthogonalizing the spatial factors X and Y brings marginal or even negative effects.

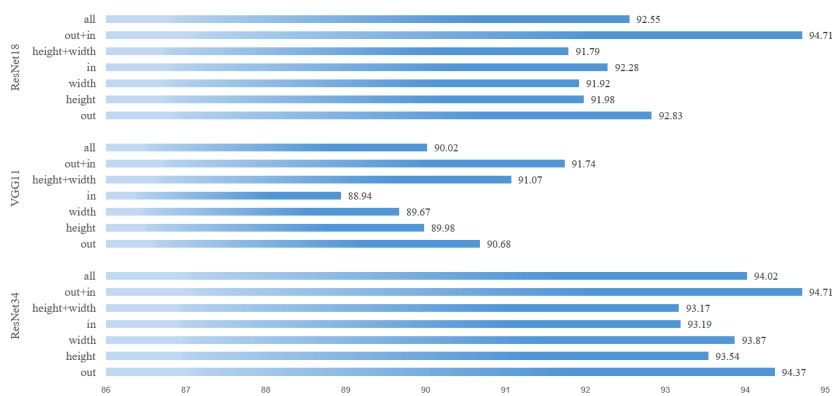

Figure 11: Comparisons between different orthogonal regularization methods.

## A.11 TIME-COST COMPARISONS BETWEEN EXISTING PEFT METHODS.

Table 15: Computational efficiencies of these PEFT methods.

|  | LoRA / LoRA+ | LAVA |
|---|---|---|
| FF | $y = (W + BA)x$ | $y = (W + BA)x$ |
| BW | $\begin{cases} \nabla_A L = B^\top(\nabla_W L) \\ \nabla_B L = (\nabla_W L)A^\top \end{cases}$ | $\begin{cases} \nabla_A L = B^\top(\nabla_W L) + 4\lambda A(A^\top A - I) \\ \nabla_B L = (\nabla_W L)A^\top \end{cases}$ |
| LC | $\mathcal{L}((W + BA)x, t)$ | $\mathcal{L}((W + BA)x, t) + \lambda\|A^\top A - I\|_F^2$ |
| TP | A, B ($2nr$) | A, B ($2nr$) |
| CC | $\mathcal{O}(Sn^2 + 2rn^2 + T)$ | $\mathcal{O}(Sn^2 + 5rn^2 + T)$ |

Table 16: Time cost between three PEFT methods.

| Method | Conv | Matrix |
|---|---|---|
| LoRA | 1.00 | 1.00 |
| LAVA | 1.20 | 1.18 |
| DoRA | 1.38 | 1.41 |

Tables 15 and 16 provide theoretical and real wall-clock time of the additional computational cost of LAVA against vanilla LoRA methods.

## A.12   VARIANCE ANALYSIS

In this section, we provide additional variance analysis of Roberta-base on the MRPC and STSB datasets from the GLUE benchmark.

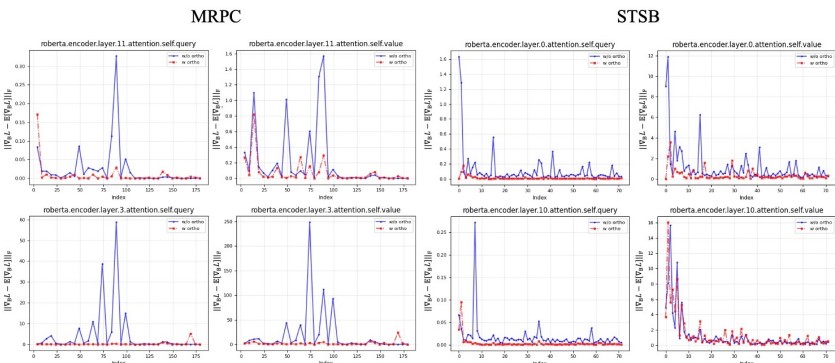

Figure 12: Variance analysis of the gradient.

## A.13   RANK ANALYSIS

In the final section, we compare the ranks between LAVA and LoRA based on the checkpoints from commonsense reasoning task. We analyzed the singular values of the increment $BA$ and give the following visualizations as shown in Figs. 13 and 14.

The LoRA updates (top row) exhibit a much smoother, almost linear decay in log-scale, with no clear spectral gap. While for LAVA, the singular values are nearly flat up to an index of about 16–17, followed by a sharp drop of several orders of magnitude. In other words, LoRA tends to keep many useless and redundant directions, hence its larger numerical rank and worse conditioning. Compared with LoRA, LAVA keeps a compact set of informative directions and strongly suppresses redundant ones, making its condition number much lower than that of LoRA, and the entire optimization better-conditioned.

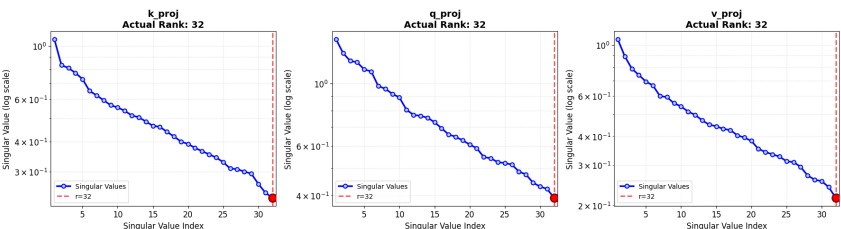

Figure 13: Singular values of LoRA.

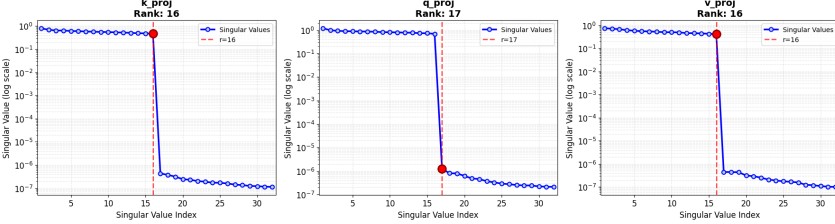

Figure 14: Singular values of LAVA.

