# OpenReview forum: "LAVA: A UNIFIED FRAMEWORK FOR FINETUNING LANGUAGE AND VISION MODELS"
_ICLR.cc/2026/Conference — Submitted to ICLR 2026_

### Official Review · Reviewer_1TBd · 2025-10-28

**Soundness:** 2
**Presentation:** 2
**Contribution:** 1
**Rating:** 2
**Confidence:** 4

**Summary:**

This paper proposes a parameter-efficient finetuning (PEFT) method named LAVA, which applies Tucker and CANDECOMP/PARAFAC decompositions to reparameterize the weights in convolutional layers. This design preserves the original weight shapes and structural properties of convolutional filters. To enhance training stability, the authors further introduce an orthogonal regularization term in the parameterization. Experiments are conducted on a variety of NLP and vision benchmarks, including language understanding, commonsense reasoning, and image segmentation, showing (limited) improvements over several baselines.

**Strengths:**

The main idea is interesting: decomposing convolutional weights while maintaining their structural form is non-trivial and practically meaningful.

Incorporating orthogonal regularization into the decomposition framework is a reasonable design choice that could potentially improve training stability and generalization.

**Weaknesses:**

1. In Equation (1), the variables *x* and *y* are not explicitly defined, which makes the formulation hard to follow.
2. The decomposition described in Section 3 is somewhat unconventional. Typically, for convolutional weights $W'$, the decomposition is applied over dimensions $c_{out}$ and $c_{in} \times h \times w$, rather than $c_{out} \times h$ and $c_{in} \times w$. The motivation and benefit of this alternative factorization are unclear, and this design choice makes the comparisons in Section 3 less convincing.
3. The experiments use **LLaMA-2-7B** as the main backbone. Given the availability of more recent models (e.g., LLaMA-3, Mistral, or Gemma), this weakens the empirical support for the claimed generality and advancement of LAVA.
4. The paper claims that orthogonal regularization improves training stability, but there are no explicit experiments or analyses (e.g., training curves or variance metrics) to support this claim.
5. The paper does not compare against several relevant and recent PEFT methods, such as **FLoRA**, **DoRA**, and **Conv-LoRA**. Tables 2 and 3 omit many state-of-the-art baselines, making it difficult to assess the true contribution and competitiveness of LAVA.

**Questions:**

See weaknesses.

---

> ### Author Response · Authors · 2025-11-25
> **Response to Reviewer 1TBd (part 1)**
>
> We sincerely appreciate the effort you’ve dedicated to providing constructive and insightful comments. We have given detailed responses below point by point, and added the discussions and additional experiments to the revised version. Please kindly let us know whether you have further concerns.
>
> # **W1**. In Equation (1), the variables *x* and *y* are not explicitly defined, which makes the formulation hard to follow.
>
> Thanks for pointing out the undefined symbols in the manuscript. In Equation (1), $x$ is the input of the current layer with the shape $R^{D}$, where $D$ is the hidden dimension of the model, and $y$ is the output of the current layer. We have corrected this in the revised version of the paper and double-checked if it has other mistakes.
>
> ---
>
> # **W2.** The decomposition described in Section 3 is somewhat unconventional. Typically, for convolutional weights $W^\prime$, the decomposition is applied over dimensions $c_{out}$ and $c_{in} \times h \times w$, rather than $c_{out} \times h$ and $c_{in} \times w$. The motivation and benefit of this alternative factorization are unclear, and this design choice makes the comparisons in Section 3 less convincing.
>
> >   Thanks the Reviewer once again because the Reviewer pointed out that the convolution layer should be reshaped into $c_{out}\times (c_{in}\times h \times w)$. We agree that this is an important factor in reshaping convolution layer.
>
> >   At the same time, our initial choice was efficiency motivated: Suppose a convolution layer with the shape $256 \times 256 \times 3 \times 3$, pre-defined rank is 8, reshaping into $(256 \times 3) \times (256 \times 3)$ would require to tune $(256 \times 3) \times 8 \times 2 = 12288$, while reshaping to $256 \times (256 \times 3 \times 3)$ require to tune $(256 \times 8)+(256\times 3 \times 3) \times 8 = 20480$ parameters, which suggests to reshape the original convolution layer into the shape $(c_{out} \times h) \times (c_{in} \times w)$. To reconcile both views, we have changed baseline implementations related to convolution layers into the new form: $c_{out}\times (c_{in}\times h \times w)$, and we guarantee that all the following baselines in the experiment, which require a reshape operation, have been re-implemented as well. **We want to point out that our reshape-free method still outperforms other baselines implemented in the new way, and we attribute this phenomenon to the reshape operation itself: whichever way it reshapes the pre-trained weight, it needs to combine dimensions together, which causes information fusion and performance degradation**. So we think that the motivation is still valid. We encourage readers to refer to **Section 3: Low-Rank Subspace Analysis** for details. All the changing parts, including data and text descriptions, are highlighted in blue.
> >
> >   To validate our method, we have compared LAVA against more baseline methods (including LoRA, VeRA, Conv-Adapter, and DoRA) in Table. 4 in the revised version.
> >   We provide the results here for convenience:
> >
> >   | Method       | params | $\delta_1 \uparrow$ | $\delta_2 \uparrow$ | $\delta_3 \uparrow$ | AbsRel $\downarrow$ | RMSE $\downarrow$ | log10 $\downarrow$ |
> >   | ------------ | ------ | ------------------- | ------------------- | ------------------- | ------------------- | ----------------- | :----------------: |
> >   | LoRA         | 2.04M  | 0.937               | 0.994               | 0.999               | 0.090               | 0.356             |       0.039        |
> >   | VeRA         | 0.57M  | 0.969               | 0.997               | 0.999               | 0.728               | 0.286             |       0.032        |
> >   | Conv-Adapter | 1.21M  | 0.969               | 0.997               | 0.999               | 0.072               | 0.278             |       0.031        |
> >   | DoRA         | 2.14M  | 0.967               | 0.997               | 0.999               | 0.074               | 0.286             |       0.032        |
> >   | LAVA | **1.14M** | **0.972** | **0.997** | 0.999 | **0.070** | 0.274 | **0.030** |
> >
>
> ---

---

> ### Author Response · Authors · 2025-11-25
> **Response to Reviewer 1TBd (part 2)**
>
> # W3. The experiments use **LLaMA-2-7B** as the main backbone. Given the availability of more recent models (e.g., LLaMA-3, Mistral, or Gemma), this weakens the empirical support for the claimed generality and advancement of LAVA.
>
> >   We have added experiments on Gemma-2-2b to compare LAVA against LoRA, VeRA, LoRA+, and DoRA in the revised version (See Table 3). The table below shows the detailed performances on each dataset, which we think could solve your question well.
>
> | Method           | BoolQ | PIQA | SIQA | HellaSwag | WinoGrande | ARC-e | ARC-c | OBQA | Avg.        |
> | ---------------- | ----- | ---- | ---- | --------- | ---------- | ----- | ----- | ---- | ----------- |
> | LoRA | 68.5 | 80.5 | 77.2 | 86.9 | 78.5   | 81.8 | 66.0 | 79.6 | 77.4       |
> | VeRA  | 65.6 | 75.4 | 74.6 | 59.3  | 72.8   | 80.0 | 63.0 | 70.6 | 70.1 |
> | LoRA+ | 68.6 | 74.8 | 77.7 | 89.4  | 77.6   | 84.1 | 66.8 | 75.0 | 77.5    |
> | DoRA  | 67.5 | 80.7 | 77.7 | 87.2   | 79.2    | 81.6 | 66.7 | 77.8 | 77.3 |
> | LAVA |68.5|80.8|77.6|88.8|80.4|83.2|66.6|78.6|78.2|
>
> ---
>
> # W4. The paper claims that orthogonal regularization improves training stability, but there are no explicit experiments or analyses (e.g., training curves or variance metrics) to support this claim.
>
> >   We have added additional variance analysis in the revised manuscript (**Subsection of How does
> orthogonal regularization affects the training dynamics in Section 6**).
> >
> >   As shown in Fig. 7, we compare the differences with and without orthogonal regularization in terms of gradient variances, and observe that without orthogonal regularization, the variance of gradients fluctuates violently. Besides, orthogonal regularization could help reduce the variance well (Green line in Fig. 7).
>
> ---
>
> # W5. The paper does not compare against several relevant and recent PEFT methods, such as **FLoRA**, **DoRA**, and **Conv-LoRA**. Tables 2 and 3 omit many state-of-the-art baselines, making it difficult to assess the true contribution and competitiveness of LAVA.
>
> > We have responded to this question previously. We encourage readers to refer to the tables shown in the first and third responses before.

---

### Official Review · Reviewer_mXqN · 2025-10-30

**Soundness:** 3
**Presentation:** 3
**Contribution:** 3
**Rating:** 4
**Confidence:** 4

**Summary:**

LAVA proposes a unified, parameter-efficient fine-tuning framework aimed at addressing two core issues in existing LoRA-based methods for vision and language tasks:

Insufficient exploration of low-rank subspace: the optimization process of LoRA may lead to redundant dimensions

Improper handling of convolution layers: flattening operations destroy spatial encoding properties

Key contributions include:

Tensor-factorization perspective: parameterizes convolutional kernel updates as a sum of rank-1 tensor components, preserving full dimensional integrity

Orthogonal regularization: theoretically shown to reduce gradient variance and stabilize training

Unified framework: applicable to both attention mechanisms and convolutional networks

**Strengths:**

Conducts the first systematic analysis of low-rank subspace properties in convolution layers within the PEFT paradigm

Provides rigorous theoretical proofs for orthogonal regularization (Theorem 1 & Proposition 1)

Establishes the mathematical connection that shows LoRA is a special case of LAVA

Covering NLU, commonsense reasoning, semantic segmentation, depth estimation, and text generation

**Weaknesses:**

The evaluation does not include several recent and competitive PEFT baselines, such as LoRA+, VeRA, DoRA, and NoRA.

The computational efficiency analysis is incomplete, as it lacks direct comparisons of training time and memory usage.

Suggestion: include comparisons with more state-of-the-art methods in the revised version.

**Questions:**

same as above

---

> ### Author Response · Authors · 2025-11-24
> **Response to Reviewer mXqN**
>
> We sincerely appreciate your time and effort in providing constructive and insightful comments. We detail our responses below point by point, and add the discussions and additional experiments to the revised version. Please kindly let us know whether you have further concerns.
>
> ---
>
> # **W1**: The evaluation does not include several recent and competitive PEFT baselines, such as LoRA+, VeRA, DoRA, and NoRA.
>
> > In the following table, we compare LAVA against LoRA, VeRA, LoRA+, and DoRA on gemma-2-2b on **commonsense reasoning** task.
>
> |  Method    |  Params (%)    |  Avg.    |
> | ---- | ---- | ---- |
> |   LoRA   |   1.07   |  77.4    |
> |   VeRA   |   0.02   |   70.2   |
> |  LoRA+    |   1.07   |   77.5   |
> | DoRA | 1.09 | 77.3 |
> | LAVA | 1.07 | **78.1** |
>
> > In the table below, we conduct more comparisons against VeRA, Conv-Adapter, and DoRA in **depth estimation** task.
>
> |  Method    |  Params    |  $\delta_1$    | $\delta_2$ | $\delta_3$ | **AbsRel** $\downarrow$ | **RMSE** $\downarrow$  | **log10**$\downarrow$ |
> | ---- | ---- | ---- | ---- | ---- | ---- | ---- | ---- |
> |   VeRA   |   0.57M   | 0.969  | 0.997 | 0.999 | 0.728 | 0.286 | 0.032 |
> |  Conv-Adapter   |  1.21M   |  0.969   | 0.997 | 0.999 | 0.072 | 0.278 | 0.031 |
> | DoRA | 2.14M | 0.967  | 0.997 | 0.999 | 0.074 | 0.286 | 0.032 |
> | LAVA | 1.14M |**0.972** | **0.997** | **0.999** | **0.070** | **0.274** | **0.030** |
>
> ---
>
> # **W2**: The computational efficiency analysis is incomplete, as it lacks direct comparisons of training time and memory usage.
>
> > We have provided additional explanations theoretically and empirically in the revised version. Firstly, we compare the total cost (including feed-forward, backward, and loss computation cost) between LoRA and LAVA **theoretically**, and the result is shown in **Table 15, in the Appendix of the revised version**. Additional orthogonal regularization would introduce $\mathcal{O}(3rn^2)$, which can be negligible at the long-sequence setting, where $r \ll S$. Here, $S$ stands for the sequence length.
>
> > Additionally, we compare the **wall-clock time** of standard LoRA, LAVA, and DoRA, as is shown in **Table 16, in the Appendix, in the revised PDF version**.  Adding orthogonal regularization would **not cost more than $\approx20\\%$** in both matrix and convolution form, which is acceptable compared with DoRA ($\approx40\\%$)
>
> > We encourage readers to refer to **Section 6, subsection: What is the additional computational overhead for orthogonal regularization?** for more details.

---

### Official Review · Reviewer_6cEr · 2025-10-31

**Soundness:** 3
**Presentation:** 3
**Contribution:** 2
**Rating:** 4
**Confidence:** 4

**Summary:**

The paper introduces LAVA (Language And Vision Adaption), a unified framework for the parameter-efficient fine-tuning of large models. The authors identify two primary limitations with the widely-used LoRA method: (1) Subspace Redundancy, where unconstrained training can lead to correlated, inefficient representations in the low-rank update, and (2) Dimension Disorder, where applying LoRA to convolutional layers requires flattening tensors into matrices, thereby disrupting the inherent spatial structure of the weights.

LAVA addresses these issues by introducing (1) a generalized subspace-based adaptation that handles high-order tensors (like convolution kernels) directly by parameterizing the weight update as a sum of rank-1 tensors, thus preserving dimensional integrity. This method naturally reduces to the LoRA formulation when applied to matrices. (2) A column-orthogonal regularization term applied to the trainable low-rank matrices. This encourages the basis vectors to be orthogonal, promoting a more complete exploration of the low-rank subspace and, as the authors show theoretically, stabilizing training by reducing the variance of gradients.

The authors conduct a comprehensive set of experiments across natural language understanding (GLUE), commonsense reasoning (LLaMA2-7B), semantic segmentation (SAM), depth estimation (Depth-Anything), and text-to-image generation (SDXL). Their results consistently show that LAVA outperforms LoRA and other PEFT baselines.

**Strengths:**

The paper is built on a very clear critique of LoRA. The concepts of dimension disorder for convolutions and dimension redundancy from unconstrained optimization are well-explained, and represent meaningful limitations in current PEFT approaches.

LAVA is a simple but effective framework that simple addresses the identified problems. Using a tensor decomposition-inspired update for convolutions is a natural fit, and extending LoRA with orthogonal regularization is a principled way to improve subspace exploration.

The empirical evaluation compares LAVA and LoRA across many tasks, further supporting the efficacy of LAVA.

**Weaknesses:**

The core components of LAVA (tensor decomposition [3,4] and orthogonal regularization) are not novel in isolation. Orthogonal constraints are a well-known tool in machine learning for improving training stability and representation quality and have been previously used for PEFT [1, 2]. The paper would be strengthened by a more detailed discussion of related work that has used similar techniques, even outside the direct context of PEFT, to better contextualize its specific contribution. For instance, the distinction from OFT could be sharpened.

The work lacks comparison (or integration ?) into more recent PEFT alternatives to LoRA [5,6,7] and others. Do these state-of-the-art PEFT approaches also suffer from subspace redundancy and dimension disorder also or are some of these problems already partly addressed. In this case, is LaVA complementary with these existing solutions ?

The text-to-image generation experiment (Sec 5.5) feels less thorough than the other experimental sections. The evaluation relies primarily on a single qualitative figure in the main paper and one FID score in the appendix. Given the stochastic nature of generation, strengthening this section with more quantitative metrics (e.g., CLIP scores), a user study, or at least more generated examples in the appendix would make the claims in this domain more robust.

[1] Xiao Wang, Tianze Chen, Qiming Ge, Han Xia, Rong Bao, Rui Zheng, Qi Zhang, Tao Gui, and Xuanjing Huang. 2023. Orthogonal Subspace Learning for Language Model Continual Learning. In EMNLP 2023.

[2] Büyükakyüz, K. OLoRA: orthonormal low-rank adaptation of large language models. arXiv preprint arXiv:2406.01775, 2024

[3] Lebedev, Vadim, et al. "Speeding-up convolutional neural networks using fine-tuned cp-decomposition." arXiv preprint arXiv:1412.6553 (2014).

[4] Yifan Yang, Jiajun Zhou, Ngai Wong, and Zheng Zhang. 2024. LoRETTA: Low-Rank Economic Tensor-Train Adaptation for Ultra-Low-Parameter Fine-Tuning of Large Language Models. In Proceedings of Association for Computational Linguistics: Human Language Technologies, 2024

[5] Edalati, Ali, et al. "KronA: Parameter-Efficient Tuning with Kronecker Adapter." Enhancing LLM Performance: Efficacy, Fine-Tuning, and Inference Techniques. Cham: Springer Nature Switzerland, 2025. 49-65.

[6] Liu, Shih-Yang, et al. "Dora: Weight-decomposed low-rank adaptation."  International Conference on Machine Learning. 2024.

[7] Albert, Paul, et al. "RandLoRA: Full-rank parameter-efficient fine-tuning of large models." ICLR (2025).

**Questions:**

The orthogonal regularization is applied to only one of the factor matrices (U in Eq. 3) in the convolutional case. What was the rationale for this specific choice? Have the authors experimented with regularizing all factor matrices or a different combination, and how did that affect performance and training stability?

Regarding the commonsense reasoning results (Table 7), do the authors have any hypotheses for why LAVA might underperform LoRA on specific datasets like SIQA and WinoGrande? Is it possible that for some tasks, the unconstrained subspace exploration of LoRA is accidentally beneficial, or is it more likely noise?

Could the authors quantify the computational overhead of the orthogonal regularization term? Does it introduce a noticeable slowdown in training wall-clock time compared to a standard LoRA implementation?

---

> ### Author Response · Authors · 2025-11-25
> **Response to Reviewer 6cEr (part 1)**
>
> We sincerely appreciate the effort you’ve dedicated to providing constructive and insightful comments. We have given detailed responses below point by point and added the discussions and additional experiments to the revised version. Please kindly let us know whether you have further concerns.
>
> # **W1**. The core components of LAVA (tensor decomposition [3,4] and orthogonal regularization) are not novel in isolation. Orthogonal constraints are a well-known tool in machine learning for improving training stability and representation quality and have been previously used for PEFT [1, 2]. The paper would be strengthened by a more detailed discussion of related work that has used similar techniques, even outside the direct context of PEFT, to better contextualize its specific contribution. For instance, the distinction from OFT could be sharpened.
>
> > RESPONSE: To address your concerns, we have discussed the differences between our method and other similar methods (i.e., **Section 2, subsection of Comparisons against existing methods**). We encourage readers to refer to this section for more details.  Here, we also give the detailed comparisons below:
>
> > **How does LAVA differ from closely related works?**
>
> > *   **OLoRA VS. LAVA**. OLoRA enforces hard orthonormality on LoRA factors **at initialization**, and then optimizes without constraint. LAVA instead uses a soft orthogonality regularizer within a CP-parameterized form, and such reparameterization could be **naturally extended to convolution form without flattening**. And we analyzed the rank effects of the regularization, empirically, we observe compact representations with improved robustness.
> > *   **OFT VS. LAVA**. OFT constrains update via orthogonal transformation. The major difference is in convolution layer: OFT **operates at full matrix level**, while LAVA's form keeps spatial structure and introduces low-rank representations, meaning that **the number of trainable parameters of LAVA is less than that of OFT**.
> > *   **LAVA VS. Lebedev, Vadim et, al[3] /  Yang et, al[4].** Classical CP factorization has been used for **compression of convolution layers**. In contrast, we use CP to **parameterize the increment**, **not to refactorize the base weights**.
>
> ---
>
> # **W2**. The work lacks comparison (or integration ?) into more recent PEFT alternatives to LoRA [5,6,7] and others. Do these state-of-the-art PEFT approaches also suffer from subspace redundancy and dimension disorder also or are some of these problems already partly addressed. In this case, is LaVA complementary with these existing solutions ?
>
> >   RESPONSE: To address your concerns, we have provided additional comparisons in the revised version (see Table. 4), in which we add comparisons against other baselines, including LoRA+, Conv-Adapter, VeRA, and DoRA. And we encourage readers to refer to Table. 4 in the revised version for details.
>
> >   What's more, we have conducted some preliminary experiments on the integration of orthogonal regularization and other baselines. Here, whether to apply orthogonal regularization is the only difference. The results are shown below:
>
> | Model      | Method           | BoolQ |     PIQA	| SIQA    |  HellaSwag    | WinoGrande | ARC-e | ARC-c | OBQA | Avg.|
> | ----- |----- | --- | --- | --- | --- |--- |--- |--- |--- |--- |
> | Gemma-2-2b | DoRA (w/o ortho) |   67.5    |   80.7   |  77.7    |  87.2    | 79.2 | 81.6 | 66.7 | 77.8 | 77.3 |
> | Gemma-2-2b | DoRA (w/ ortho) | 69.3 | 80.8 | 78.4 | 87.1 |78.4|81.6|67.7|76.8|77.5 (+0.2)|
> | Llama3-2-1b | DoRA (w/o ortho) | 63.5 | 74.8 | 71.2 | 47.2 |68.3|68.7|52.5|66.0|64.0|
> | Llama3-2-1b |DoRA (w/ ortho)|61.8|75.8|72.4|65.5|68.0|69.7|53.1|68.4|66.8 (+2.8)|
>
> >   Orthogonal regularization seems to be a plug-and-play module to existing PEFT methods, and we hypothesize that existing PEFT methods suffer from dimension redundancy as well. However, due to time and resource constraints, we will leave this for future work.

---

> > ### Author Response · Authors · 2025-11-25
> > **Response to Reviewer 6cEr**
> >
> > # **W3**. The text-to-image generation experiment (Sec 5.5) feels less thorough than the other experimental sections. The evaluation relies primarily on a single qualitative figure in the main paper and one FID score in the appendix. Given the stochastic nature of generation, strengthening this section with more quantitative metrics (e.g., CLIP scores), a user study, or at least more generated examples in the appendix would make the claims in this domain more robust.
> >
> > >    We have provided additional metric scores of the generated images and more generated examples from the text-to-image generation experiment in the revised version in the Appendix(See Table 14 and Fig. 10 for details). Besides, from Fig.10, we observe that images generated from LAVA show softer colors and smoother colors, which we believe is more in line with the style of training images. We encourage readers to refer to **Section A.9: Text-to-image Generation Experiment Details** for additional information. We also provide the metric scores below for convenience:
> >
> > | Metric                 | LoRA   | LAVA   |
> > | ---------------------- | ------ | ------ |
> > | FID Score $\downarrow$ | 480.41 | 429.48 |
> > | CLIP Score $\uparrow$  | 0.57   | 0.60   |
> >
> >
> >
> > ---
> >
> > # **Q1**. The orthogonal regularization is applied to only one of the factor matrices (U in Eq. 3) in the convolutional case. What was the rationale for this specific choice? Have the authors experimented with regularizing all factor matrices or a different combination, and how did that affect performance and training stability?
> >
> > >   Applying orthogonal regularization on a convolution layer with the shape $c_{out}\times c_{in}\times w\times h$ has many options, and we have analyzed the influences of different regularization methods and presented the results in the revised version (See Section A.10 and Fig. 11 for details). Applying orthogonalization on both input and output channels (U and V in Eq. 3) achieves the best result, followed by the method of regularizing only on output channel (i.e., U in Eq. 3). We believe that such phenomenon is in line with our intuition: orthogonality on input and output channels allows every filter in the convolution layer learns different representations, and the relation between different filters is reduced, making the optimization of each filter easier. Considering the accuracy and efficiency, we decided to regularize on $U$ to achieve competitive performances.
> >
> > ---
> >
> > # **Q2**. Regarding the commonsense reasoning results (Table 7), do the authors have any hypotheses for why LAVA might underperform LoRA on specific datasets like SIQA and WinoGrande? Is it possible that for some tasks, the unconstrained subspace exploration of LoRA is accidentally beneficial, or is it more likely noise?
> >
> > >   Thanks for pointing out this observation, and we believe this is an interesting question. Below, we give our hypothesis as a response.
> >
> > >   LAVA adds an explicit orthogonality constraint on the low-rank basis, which can be viewed as selecting a well-conditioned canonical representative within the equivalence class of LoRA factorizations $BA$. This improves optimization geometry and typically yields more stable feature updates. We analyzed the ranks of the learned matrices from different PEFT methods in Section A.13: Rank Analysis in the Appendix, where we visualized that the effective rank of commonsense reasoning may be 16-17, while LoRA uses the entire 32 dimensions for optimization. The redundant dimensions are more likely to learn the noise from the data, making the performance unstable. We hope this can explain the underperformance of LAVA on some specific datasets.
> >
> > ---
> >
> > # **Q3**. Could the authors quantify the computational overhead of the orthogonal regularization term? Does it introduce a noticeable slowdown in training wall-clock time compared to a standard LoRA implementation?
> >
> > >   We have provided both theoretical and empirical results of the additional computational overhead from orthogonal regularization in the revised version. In theory, the additional overhead is $\mathcal{O}(3rn^2)$, which is negligible in the long-sequence setting, as $r\ll S$. We also counted the wall-clock time of LoRA, LAVA, and DoRA. It turns out that LAVA costs an additional ~20% compared with standard LoRA in both matrix and convolution form, while DoRA has nearly 40% additional clock overhead. We encourage readers to refer to **Section 6: What is the additional computational overhead for orthogonal regularization?** for more details.

---

### Official Review · Reviewer_R3Bj · 2025-11-02

**Soundness:** 2
**Presentation:** 3
**Contribution:** 2
**Rating:** 6
**Confidence:** 4

**Summary:**

The work proposes a method (LAVA) for parameter efficient fine-tuning of vision and language models, particularly focused on convolution layers. Given a tensor, it models the low-rank update as the sum of 'r' CANDECOMP/PARAFAC (CP) rank-1 updates. Each rank-1 update is obtained as an outer-product of learnable vectors. When the tensor is a matrix, this is equivalent to LoRA. Unlike typical application of LoRA to convolution layers by reshaping the weight tensor to be a matrix, the proposed CP rank-1 update in LAVA does not require any reshaping and thus better models spatial information. Additionally, the authors propose orthogonal regularization to ensure the update has maximum possible rank and to improve training stability. The proposed method is shown to outperform baseline approaches including LoRA on diverse vision and language tasks.

**Strengths:**

1. The idea of using the sum of CP rank-1 updates for convolution layers is interesting. It addresses the specific issue of weight reshaping in LoRA adaptation for convolution.
2. The idea of orthogonal regularization is well motivated. The authors provide theoretical proof for training stabilization due to the proposed regularization.
3. The experiments include diverse tasks on both vision and language models and the proposed method consistently outperforms both LoRA and other baseline approaches.

**Weaknesses:**

1. The primary contribution of the paper is an effective PEFT method (LAVA) for convolutional layers. However, there is not much discussion or empirical comparison with related PEFT methods focused on convolution like Lora-C [a], Conv-Adapter [b] and LoRAE [c]. Both Conv-Adapter and LoRAE preserve spatial properties in convolution similar to LAVA. Conv-Adapter can be seen as a generalization of LAVA and reduces to LAVA when the learnable 2-D filter is modified to be a separable filter and removing the non-linearity between the depth-wise and point-wise convolution blocks. The experiments are limited with just results with a single dataset and model on depth estimation and image generation tasks. There is no comparison with SOTA approaches on the image generation task. The results on semantic segmentation (table 12 in A.7.2) are not consistent with the existing literature (LoRA consistently outperforms Conv-Lora while Conv-Lora (Zhong et al., 2024) show the opposite on the same datasets).
2. For language models, the proposed approach reduces to applying orthogonal regularization atop LoRA. More experimental results and analysis is required to understand whether and why this is helpful. For instance, does the regularization lead to a higher rank than that observed in LoRA? Or, is the stabler training the reason for performance improvements? The provided LLM results are on just two datasets with just one model (small RoBERTa model on one, LLaMA-2 on another) for each. While I understand the resource requirements for larger scale experiments, more experiments are required for a stronger comparison between LoRA and the proposed method. Discussion and comparison with a related work OLoRA [d] is missing. OLoRA performs orthonormal decomposition of the weight matrix before performing LoRA updates.

References:

[a] Ding, Chuntao, et al. "LoRA-C: Parameter-Efficient Fine-Tuning of Robust CNN for IoT Devices." arXiv preprint arXiv:2410.16954 (2024). \
[b] Chen, Hao, et al. "Conv-adapter: Exploring parameter efficient transfer learning for convnets." Proceedings of the IEEE/CVF conference on computer vision and pattern recognition. 2024. \
[c] Wang, Zhixue, Hongyao Ma, and Jiahui Zhai. "Low-rank adaptation for edge AI." Scientific Reports 15.1 (2025): 33109. \
[d] Büyükakyüz, Kerim. "Olora: Orthonormal low-rank adaptation of large language models." arXiv preprint arXiv:2406.01775 (2024).

**Questions:**

1. Provide analysis on the rank of learned weight matrices for the language model experiments for both LoRA and LAVA.
2. The learning rate plays a very important role in LoRA fine-tuning. Since the learning rate is tuned for a particular model and dataset in the 3. LLM experiments, the comparison with LoRA might be unfair. Provide results on language modeling with lr tuning for LoRA. Also, provide details on the dataset split used to perform the lr tuning for LAVA.
4. Provide empirical comparison with Conv-adapter either on existing tasks in LAVA or on the tasks in [b].
5. Provide discussion on training compute and memory for LAVA. For LLMs, does splitting the update into `r` matrices significantly increase the training compute and memory compared to LoRA? How does this scale to larger models and ranks?
Add results for multiplier=0 for the plots in Figure 6 (analysis of \lambda). The value of \lambda does not seem to significantly affect results on the NLU tasks and a multiplier of value lower than 1 seems to have the best performance. Why would LAVA then perform significantly better than LoRA?
6. Why are the results for LoRA (encoder+decoder) so much worse than LoRA (encoder) in depth estimation (table 3)? Is it because of the convolutional decoder? Why do we not observe similar degradation in segmentation and image generation tasks? Does this support the use of non-reshaping technique in LAVA? More such experiments on convolution heavy backbones like ResNet would have made the work stronger (not asking for those expts here).

---

> ### Author Response · Authors · 2025-11-25
> **Response to Reviewer R3Bj (Part 1)**
>
> We sincerely appreciate the effort you’ve dedicated to providing constructive and insightful comments. We have given detailed responses below point by point, and added the discussions and additional experiments to the revised manuscript. Please kindly let us know whether you have further concerns.
>
> # W1. The primary contribution of the paper is an effective PEFT method (LAVA) for convolutional layers. However, there is not much discussion or empirical comparison with related PEFT methods focused on convolution like Lora-C, Conv-Adapter and LoRAE. Both Conv-Adapter and LoRAE preserve spatial properties in convolution similar to LAVA. Conv-Adapter can be seen as a generalization of LAVA and reduces to LAVA when the learnable 2-D filter is modified to be a separable filter and removing the non-linearity between the depth-wise and point-wise convolution blocks. The experiments are limited with just results with a single dataset and model on depth estimation and image generation tasks. There is no comparison with SOTA approaches on the image generation task. The results on semantic segmentation (table 12 in A.7.2) are not consistent with the existing literature (LoRA consistently outperforms Conv-Lora while Conv-Lora show the opposite on the same datasets).
>
> > RESPONSE: To address your concerns, we tested Conv-LoRA using the author's public repo. We can find that Conv-LoRA is not deterministic in implementation, making the results less reproducible and less convincing. We have changed the implementation into a deterministic way, and reported the average results in Table 12 in Appendix A7.2 (Table 13 in the revised manuscript) using random seeds 42, 43, and 44.
>
> # W2. For language models, the proposed approach reduces to applying orthogonal regularization atop LoRA. More experimental results and analysis is required to understand whether and why this is helpful. For instance, does the regularization lead to a higher rank than that observed in LoRA? Or, is the stabler training the reason for performance improvements? The provided LLM results are on just two datasets with just one model (small RoBERTa model on one, LLaMA-2 on another) for each. While I understand the resource requirements for larger scale experiments, more experiments are required for a stronger comparison between LoRA and the proposed method. Discussion and comparison with a related work OLoRA [d] is missing. OLoRA performs orthonormal decomposition of the weight matrix before performing LoRA updates.
>
> >    RESPONSE: To address your concern, we have compared the proposed method  LoRA, VeRA, LoRA+, and DoRA on Gemma-2-2b in the commonsense reasoning task (Table 1 below / or Table 3 in the revised manuscript), and VeRA, Conv-Adapter, and DoRA on the depth-estimation task (Table 2 below or Table 4 in the revised manuscript).
>
> Table 1: Performance comparisons of Gemma-2-2b on eight commonsense reasoning datasets.
>
> | Method | Params (%) | Avg. |
> | ------ | ---------- | ---- |
> | LoRA   | 1.07       | 77.4 |
> | VeRA   | 0.02       | 70.2 |
> | LoRA+  | 1.07       | 77.5 |
> | DoRA   | 1.09       | 77.3 |
> | LAVA   | 1.07       | 78.1 |
>
> Table 2: Comparison of different methods for finetuning Depth-Anything.
> | Method       | params    | $\delta_1 \uparrow$ | $\delta_2 \uparrow$ | $\delta_3 \uparrow$ | AbsRel $\downarrow$ | RMSE $\downarrow$ | log10 $\downarrow$ |
> | ----- | ---- | ---- | ----- | ----- | ----- | ---- | :-------: |
> | LoRA         | 2.04M     | 0.937               | 0.994               | 0.999               | 0.090               | 0.356             |       0.039        |
> | VeRA         | 0.57M     | 0.969               | 0.997               | 0.999               | 0.728               | 0.286             |       0.032        |
> | Conv-Adapter | 1.21M     | 0.969               | 0.997               | 0.999               | 0.072               | 0.278             |       0.031        |
> | DoRA         | 2.14M     | 0.967               | 0.997               | 0.999               | 0.074               | 0.286             |       0.032        |
> | LAVA         | **1.14M** | **0.972**           | **0.997**           | 0.999               | **0.070**           | 0.274             |     **0.030**      |
>
> >   We have also mentioned the difference between our method and OLoRA, as well as other related methods,  in the **Section of Related Works**. We also encourage readers to refer to **Section 2, subsection: Comparisons against existing methods** in the revised manuscript for details.
>
> >   OLoRA proposes to initialize A and B using the first r QR-decomposition components of the pre-trained matrix to speed up convergence, which means that it initializes A by an orthonormal matrix. LAVA focuses on training dynamics and approximates A as an orthogonal matrix during training. However, OLoRA cares about initialization and only thinks of the first step of optimization. We believe that our method is based on a completely different idea from OLoRA.

---

> ### Author Response · Authors · 2025-11-25
> **Response to Reviewer R3Bj (Part 2)**
>
> # **Q1**. Provide analysis on the rank of learned weight matrices for the language model experiments for both LoRA and LAVA.
>
> >   RESPONSE: To address your concerns, we analyzed the checkpoints from the commonsense reasoning task, and then conducted SVD operation on the increment of the pre-trained model (i.e., BA in LoRA and LAVA settings). We plotted singular values in descending order, and the experimental results can be found in the revised manuscript (See Figs. 13 and 14). In Fig. 13, LoRA exhibits a much smoother, almost linear decay in log-scale, meaning that the energy is evenly distributed among these dimensions. It tends to use the entire space for optimization. While for LAVA (Fig. 14), the singular values are nearly flat up to an index of about 16-17, followed by a sharp drop of several orders of magnitude, meaning that setting a pre-defined rank to be at 16-17 is enough for downstream tasks.
>
> >   Based on the observation above, LoRA has many redundant dimensions during optimization. The additional dimensions could fit noise during training, only to reduce the model’s generalization ability. While for LAVA, it keeps a compact set of informative directions and strongly suppresses redundant directions, making the entire optimization better conditioned.
>
> ---
>
> # **Q2.** The learning rate plays a very important role in LoRA fine-tuning. Since the learning rate is tuned for a particular model and dataset in the 3. LLM experiments, the comparison with LoRA might be unfair. Provide results on language modeling with lr tuning for LoRA. Also, provide details on the dataset split used to perform the lr tuning for LAVA.
>
> >   RESPONSE: for NLU tasks, the results of all baselines are taken from [1] and [2], and the detailed dataset split is shown in Table 6 in the revised manuscript. We follow the same setting as [1] and [2] to perform lr tuning for LAVA.
>
> >   For the commonsense reasoning tasks, the results of baseline methods and dataset split are taken from [3].
>
> [1]: Muling Wu, Wenhao Liu, et al. Advancing parameter efficiency in fine-tuning via representation editing. Proceedings of the 62nd Annual Meeting of the Association for Computational Linguistics.
>
> [2]: Zhengxuan Wu, Aryaman Arora, et al. Reft: Representation finetuning for language models, 2024. (arXiv:2404.03592).
>
> [3]: Zhiqiang Hu, Lei Wang, et al. LLM-adapters: An adapter family for parameter-efficient fine-tuning of large language models. (EMNLP 2023).
>
> ---
>
> # **Q3**. Provide empirical comparison with Conv-adapter either on existing tasks in LAVA or on the tasks in [b].
>
> >   RESPONSE: To address your concerns, we have provided additional comparisons in the revised manuscript (see Table 4), in which we add comparisons against other baselines, including Conv-Adapter. And we encourage readers to refer to Table 4 in the revised version for details.
>
> ---
>
>  # **Q4 (a)**. Provide discussion on training compute and memory for LAVA. For LLMs, does splitting the update into r matrices significantly increase the training compute and memory compared to LoRA? How does this scale to larger models and ranks?
>
> >   RESPONSE: To address your concerns, we have provided additional discussions about the training computation and memory cost in the revised version (i.e., **Subsection: What is the additional computational overhead for orthogonal
> regularization?**). These are experiments conducted at rank=32, which we believe is a common setting when finetuning LLMs (the rank is often set between 4 to 32, and merely set even larger). As shown in Table 15, the computational overhead is negligible in theory in the long-sequence setting. From Table 16, we find that the actual overhead is around 20% in both matrix and convolution form, which is acceptable compared with DoRA (~40%).
> >
>
> # **Q4(b)**. Add results for multiplier=0 for the plots in Figure 6 (analysis of \lambda). The value of \lambda does not seem to significantly affect results on the NLU tasks and a multiplier of value lower than 1 seems to have the best performance. Why would LAVA then perform significantly better than LoRA?
>
> >   RESPONSE: To address your concerns, we have added results for the case $\lambda=0$ in the revised manuscript (See Fig. 6).
>
> >   When conducting NLU experiments, the number of epochs is set to be large (STS-B: 40, QNLI: 25, QQP: 25), which guarantees the convergence of the model. We believe that the important thing is not about what the real value of $\lambda$ (the value of $\lambda$ indeed decides the convergence rate of matrix A towards orthogonal, but typically after no more than 1 epoch, the orthogonal loss of A becomes small enough based on our observation), instead, **what helps is the presence of orthogonality, not a large coefficient**. Enforcing near-orthogonal columns in A decouples the rank components, and reduces redundant directions (See Section A.13: Rank Analysis in the appendix of the revised version for details), making the optimization better-conditioned.

---

> ### Author Response · Authors · 2025-11-25
> **Response to Reviewer R3Bj (Part 3)**
>
> # **Q5.** Why are the results for LoRA (encoder+decoder) so much worse than LoRA (encoder) in depth estimation (table 3)? Is it because of the convolutional decoder? Why do we not observe similar degradation in segmentation and image generation tasks? Does this support the use of non-reshaping technique in LAVA? More such experiments on convolution heavy backbones like ResNet would have made the work stronger (not asking for those expts here).
>
> > When reshaping convolution layers with the shape $c^{out}\times c^{in}\times w \times h$ , there are many forms to flatten the convolution layer into a matrix, as discussed in our response to Reviewer 1TBd. We encourage readers to refer to the discussion there for more details, including why we designed to do so at the beginning and the following changes in the experiment outcomes. After using the new reshape way, the significant performance drop disappears, but LAVA still performs better against other baselines. We want to emphasize here that it is the reshape operation that really matters in performance degradation, instead of the way we reshape this. Thus, we think our finding is valid, and LAVA’s outstanding performance is consistent. We hope this will explain our thinking well. Lastly, we want to apologize here for all the misleading results in the original version.

---

### Meta-Review · Area_Chair_PcFQ · 2026-01-09

**Summary:**

It received scores of 6,4,4,2. The authors submitted a rebuttal but reviewers did not get the chance to reply.

**Reviewer Concerns:**

The main concerns are: missing and inadequate comparisons with recent PEFT methods, the core components are not novel in isolation, use of limited datasets in experiments, and incomplete efficiency analysis. The rebuttal addresses some of the concerns. For instance, it compares with some recent PEFT methods. However, I do not think it would have changed the scores much to change the final decision to accept.

**Reviewer Scores:**

6,4,4,2. I do not think the reviewers would have changed their score since the core concerns like limited novelty are not convincingly addressed.

---

### Decision · Program_Chairs · 2026-01-26

Reject